# Characterisation and surface radiative impact of Arctic low clouds from the IAOOS field experiment

Julia Maillard[1], François Ravetta[1], Jean-Christophe Raut[1], Vincent Mariage[1], and Jacques Pelon[1]

[1]LATMOS/IPSL, Sorbonne Université, UVSQ, CNRS, Paris, France

**Correspondence:** Julia Maillard (julia.maillard@latmos.ipsl.fr)

**Abstract.** The Ice, Atmosphere, Arctic Ocean Observing System (IAOOS) field experiment took place from 2014 to 2019. Over this period, more than 20 instrumented buoys were deployed at the North Pole. Once locked into the ice, the buoys drifted for periods of a month to more than a year. Some of these buoys were equipped with 808 nm wavelength lidars which acquired a total of 1777 profiles over the course of the campaign. This IAOOS lidar dataset is exploited to establish a novel statistic of cloud cover and of the geometrical and optical characteristics of the lowest cloud layer. The average cloud frequency from April to December over the course of the campaign was 75%. Cloud occurrence frequencies were above 85% from May to October. Single layers are thickest in October/November and thinnest in the summer. Meanwhile, their optical depth is maximum in October. On the whole, the cloud base height is very low, with the great majority of first layer bases beneath 120 m. In April and October, surface temperatures are markedly warmer when the IAOOS profile contains at least one low cloud than when it does not. This temperature difference is statistically insignificant in the summer months. Indeed, summer clouds have a shortwave cooling effect which can reach $-60$ W m$^{-2}$ and balance out their longwave warming effect.

## 1 Introduction

The Arctic is a key region of climate change: it is warming about twice as fast as the middle latitudes. This phenomenon, called "Arctic amplification", is most commonly attributed to the ice-albedo feedback, which is due to areas of open ocean exposed by melting sea ice absorbing more solar radiation. However some models with fixed albedos also appear to show amplified warming in the Arctic, pointing to other mechanisms at work (Winton, 2006; Pithan and Mauritsen, 2014). Clouds are one of the main contributors to uncertainty in global climate models because cloud feedbacks and cloud-aerosol interactions are still poorly understood; however, clouds appear to be of particular importance in the Arctic (Tjernström et al., 2008), where they play a very important role in the climate system. Indeed, Arctic clouds are observed to influence the melting of sea ice (Kay and Gettelman, 2009) and may exert control on the ice-albedo feedback this way. However, these effects and processes are seasonally variable and not well represented by annual means (Kay and Gettelman, 2009).

Firstly, the cloud cover in the Arctic has a large seasonal variability: it is especially extensive in the summer and reaches a minimum in the winter (Curry et al., 1988, 1996). This result is well attested in the literature although values and trends tend to differ between studies and instruments. For example, during the Surface Heat Balance of the Arctic (SHEBA) campaign, winter cloud occurrence measured from a combined radar/lidar was 70%. It increased to over 80% in the summer months and reached

a 95% peak in September (Shupe et al., 2006). Using data from CALIPSO (Cloud-Aerosol Lidar and Infrared Pathfinder Satellite Observations), Zygmuntowska et al. (2012) find two peaks of 85% and 90% in May and October respectively, and a minimum in January-March around 70%, in good agreement with Shupe et al. (2006). However, in the same study, cloud fractions retrieved from the space-borne Advanced Very-High-Resolution Radiometer (AVHRR) instrument were < 60% for the whole October-April period, and never rose above 80%.

Cloud microphysical characteristics and radiative impact are also seasonally-dependant. Winter clouds contain mostly ice and are therefore less emissive than summer liquid-containing clouds, although mixed-phased clouds maintain themselves throughout the year (Morrison et al., 2011). However, seasonal statistics of cloud optical depth (COD) over the Arctic ocean are scarce and uncertain: based on the AVHRR radiometer data for example, Wang and Key (2004) found a slight seasonal variation in the cloud optical depth over the Arctic ocean, with a peak in May and October (> 6) and lower values ($\approx 5$) in the winter. It has been shown that cloud radiative forcing is positive (i.e., clouds warm the surface) for much of the year, except for a short period in late June to early July when the cloud shortwave forcing is larger than the longwave forcing (Intrieri et al., 2002a). Indeed, in contrast to winter, clouds impact the surface radiative budget in two competing ways in the summer. As in winter, they provide longwave warming; but they also have a shortwave cooling effect, by preventing solar radiation from reaching the surface.

Large uncertainties remain about the characteristics of Arctic clouds and their surface impact, in part because more data and observations are needed (Kay et al., 2016). Ground-based measurements are sparse in the Arctic because of the harsh conditions and the lack of permanent settlements. The ground-based measurement stations of the International Arctic Systems for Observing the Atmosphere (IASOA) network (Uttal et al., 2016), for example Eureka (Nunavut, Canada) or Barrow (Alaska) are necessarily coastal. Nevertheless, ground-based stations have continuous data coverage with a record covering several years, and have therefore given precious information on Arctic clouds and their properties (Shupe et al., 2011; Nomokonova et al., 2019). Measurements on the sea-ice take the form of ship-based or airborne campaigns, covering only a narrow spatial and temporal window. The first such campaign was SHEBA, which covered a full year from October 1997 to October 1998. Although it yielded significant results (Stramler et al., 2011; Shupe et al., 2006), it is now more than 20 years old and not representative of the modern Arctic. Subsequent campaigns aimed at studying the Arctic's changing conditions such as the Arctic Summer Cloud Ocean Study (ASCOS) (Tjernström et al., 2014), the ACLOUD/PASCAL campaign (Wendisch et al., 2019), the Arctic Clouds in Summer Experiment (ASCE) (Sotiropoulou et al., 2016) or the Norwegian Young Sea Ice Experiment (N-ICE) (Walden et al.) covered one to six months, disproportionately in the summer. Most recently, the Multidisciplinary drifting Observatory for the Study of Arctic Climate (MOSAiC) campaign is a one year-long study of the Arctic climate, with clouds as one of many research axes. The drift is due to end in September 2020.

In this context, many established statistics - e.g., Wang and Key (2004) - make use of satellite measurements, which have large coverage but are flawed at high latitudes. Indeed, spectroradiometers (such as MODIS, or the AVHRR) may have difficulties in distinguishing clouds from the underlying sea-ice. Their performance also differs between the dark winter months and the summer (Zygmuntowska et al., 2012). All in all, there are large differences in measured values between instruments (Chan and Comiso, 2013). Satellite-based lidars such as the instrument aboard CALIPSO give more reliable measurements but

are limited to 82°N because of the satellite flight path (Winker et al., 2009). Their record is also more limited in time than that of ground-based stations (from 2006 for CALIPSO).

This paper presents results of the Ice, Atmosphere, Arctic Ocean Observing System (IAOOS) field experiment lidar measurements. This novel database offers a ground-based view of lower tropospheric clouds at very high latitudes (over 80°N) over a significant period of time - from 2014 to 2019 (Mariage, 2015). A small part of this dataset has already been analysed in Di Biagio et al. (2018) and Mariage et al. (2017). Here it is treated as a whole to extract a multi year statistic of the April to December cloud cover along the track of the drifting buoys. First, the IAOOS field campaign and other relevant datasets are presented (Sect. 2). Then the treatment of the IAOOS lidar data and the derivation of cloud characteristics are explained (Sect. 3). The obtained statistics of cloud frequency, and geometrical and optical properties are presented in Sect. 4. Finally the impact of clouds on surface temperatures and radiative balance is explored (Sect. 5).

## 2 Data used

### 2.1 The IAOOS field campaign: a 5 year study of the Arctic troposphere

#### 2.1.1 Deployed instruments

The IAOOS field experiment was led by Sorbonne University - through the LATMOS and LOCEAN laboratories - with the support of several structures, among which the French polar institute IPEV (Institut polaire français Paul-Emile Victor) and the technical division of the Institute for Earth Sciences and Astronomy (CNRS-INSU) from 2014 to 2019. The main campaign objective was to "collect real time observations of the ocean, ice, snow and atmosphere of the Arctic", offering a complementary viewpoint to that of satellites (L2). In order to do this, several instruments were installed on an autonomous floating platform (or buoy). These buoys were then locked into the pack ice and left to drift with it for a duration of several months to a year. During that time period, the buoys were tracked by GPS and communicated the acquired data to the IPEV office in Brest (48°23′24″ N, 4°29′24″ W) every day.

The main instrument on the "atmosphere" side of the buoys was a micro lidar, which was was designed to study lower troposphere and has a clear-sky range of around $4.4$ km in the daytime, and $13.7$ km at night, with a vertical resolution of $15$ m (Mariage, 2015; Mariage et al., 2017). The wavelength was chosen in the near infrared ($808$ nm) in order to avoid disturbing the local fauna while maintaining a distinct molecular signal. This is similar to many commercial ceilometers (Mariage, 2015). However, it had to be custom made to resist the tough Arctic conditions. Indeed, several key components of a lidar are sensitive to ambient temperature variations, and the buoys' operating conditions in the pack ice could be up to $40$°C colder than the lab where it was calibrated. The lidar therefore had to be modified and isolated in order to keep it at a near constant temperature (Mariage, 2015). Furthermore, the tube containing the lidar emitter and receiver was topped with a window that, in operating conditions, was often covered by frost. This layer of frost attenuates the signal, and, in extreme cases, totally blinds the lidar. In order to overcome this problem a window heating system was put in place. The actual heating was limited to the 10-minute interval before the two- to four-time daily profile acquisition in order to avoid draining the battery too fast. Theoretically, this

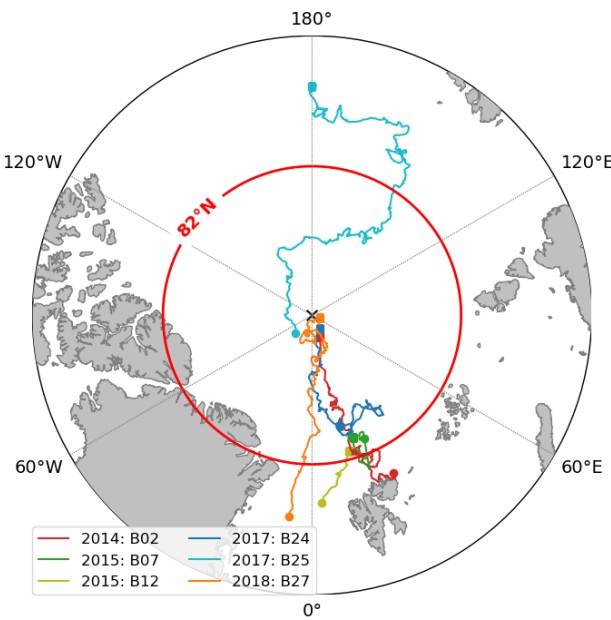

**Figure 1.** Map of the IAOOS buoy tracks, 2014-2019 (this map only includes buoys which delivered the lidar data exploited in this article). The different colours correspond to the different buoys, with the year of launch indicated. The red circle corresponds to the 82°N latitude: north of this circle, no satellite lidar data is available.

ensured that the lidar window was clear during measurement. However, in practice, the frost prevented lidar measurements from mid-December to early March. The frost problem will be further detailed in Sect. 3.1.1.

The buoys were also equipped with temperature and pressure sensors for measuring outside conditions; and internal temperature and humidity sensors for monitoring the lidar system. On the underwater portion of the buoys, a float measured ocean temperature and salinity while an Ice Mass Balance system acquired temperature profiles of the snow, ice and liquid water layers - see Koenig et al. (2016).

### 2.1.2   Buoys and tracks

The first IAOOS platform was deployed in 2013. Since then, more than 20 buoys have drifted in the Arctic pack ice, and the last one was deployed in August 2019. However, not all buoys were equipped with lidars and not all deployed lidars operated successfully. In particular, the data transmission system of the 2016 buoys functioned poorly, and there are no exploitable lidar profiles from July 2015 to March 2017 (see Table 1). All in all, five buoys yielded usable lidar data, amounting to 1777 profiles covering the April to December months. A vast majority of the drift took place north of 82°N (red circle, Fig. 1). Furthermore,

apart from one buoy, all trajectories were confined to the Atlantic sector of the Arctic, reflecting the transpolar drift stream. Indeed, most buoys studied here were locked into the ice close to the North Pole.

| Buoy | Start date | End date | Nb of exploitable profiles |
|------|-----------|----------|----------------------------|
| B02 | 13/04/2013 | 02/12/2014 | 462 |
| B12 | 26/04/2015 | 05/06/2015 | 73 |
| B24 | 06/04/2017 | 20/11/2017 | 322 |
| B25 | 15/08/2017 | 28/10/2018 | 429 |
| B27 | 19/04/2018 | 17/03/2019 | 491 |

**Table 1.** Start and end date of the buoy lidar data acquisition and number of exploitable profiles. Note that buoy B07 also yielded some profiles (Di Biagio et al., 2018) which are not treated here.

## 2.2 Other data

### 2.2.1 N-ICE

The Norwegian Young Sea Ice Experiment (N-ICE) campaign took place from January to June 2015. During that time, the
research vessel Lance drifted with four different ice floes (Walden et al.; Cohen et al., 2017; Walden et al., 2017). The first two drifts took place during the winter (January - March 2015) while the last two drifts occurred in the late spring to early summer period (April to June 2015). On each floe, a "Supersite" ice camp was installed about 300 m away from the research vessel. Atmospheric measurements were mostly performed at this Supersite. Surface longwave fluxes (up and down) were measured with a Kipp & Zonen CGR4 pyrgeometer, which has a $4.5$ to $42$ $\mu$m bandwith. The shortwave fluxes (up and down) were
measured with a Kipp & Zonen CMP22 pyranometer (200 to 3600 nm bandwidth). Both these instruments were heated and ventilated using a Kipp & Zonen CVF4 unit. Their accuracy is $3\%$ (or 5 W m$^{-2}$) for the shortwave, and $2\%$ (or 3 W m$^{-2}$) for the longwave (Walden et al., 2017; Hudson et al., 2016). The temperature at two meters was measured with a ventilated and shielded Vaisala HMP-155A sensor which has an accuracy of $2.4\%$ (or 0.3°C) (Graham et al., 2017; Cohen et al., 2017). In addition, radiosondes were launched twice-daily from the research vessel, yielding profiles of relative humidity, temperature
and wind speed (Walden et al., 2017).

Four IAOOS buoys were deployed during this campaign and drifted in the ice floe close to the research vessel. In particular, the B12 buoy was locked into the third ice floe 200m away from the Supersite from end of April to the beginning of June 2015 (Fig. 1). Because of the proximity of the buoy to the Supersite over this period, the N-ICE surface radiative flux and temperature measurements can be used as a complement to the IAOOS data. This allowed us to evaluate the radiative impact
of clouds on the surface in late spring to early summer (Sect. 5.2.1 and 5.3).

### 2.2.2 ERA5

ERA5 is the new reanalysis from the European Center for Medium-Range Weather Forecast, replacing ERA-Interim (Hersbach et al., 2020). ERA5 provides hourly or four times daily estimates of many weather variables on a 0.25°x0.25° grid and with 137 vertical levels. It is made available online with a three month delay (Copernicus Climate Change Service (C3S), 2017). Here we interpolated the ERA5 values on the IAOOS positions using bilinear interpolation in space (and linear interpolation in time) during the N-ICE drift period. This allowed us to compare the radiative flux values measured during N-ICE with the ERA5 reanalyses (see Sect. 5.2.1).

## 3 Methodology of the IAOOS lidar data treatment

### 3.1 Overcoming Arctic-specific challenges

#### 3.1.1 Lidar window frost

Several problems are associated with the autonomous drift of a lidar in harsh Arctic conditions, as outlined in Sect. 2.1. In particular, the cold conditions cause frost to form on the lidar window, because the installed window heating system could not operate the whole time in order to preserve batteries. This caused the signal to be attenuated and therefore the system constant $C$ - which is the ratio of the raw signal in photon numbers to the actual signal - to diminish.

Because it is crucial to know the system constant value in order to extract geophysical information from the raw lidar signal, this effect had to be corrected. The correction method was put in place by Mariage (2015). First a "frost index", $\gamma$ is defined:

$$\gamma = \frac{P_0}{P}$$

where $P$ is the lidar window reflection peak, and $P_0$ the minimal value taken by $P$ over the course of a drift. $P_0$ is therefore assumed to be the value of the reflection peak when the window is entirely frost-free. $\gamma$ then ranges from approximately 1 when the window is frost free to very low values ($< 5 \cdot 10^{-2}$) when the window is totally opaque. In fact, this frost index becomes a proxy for the window transmittance.

Under the assumption that aerosol load is very low in the high Arctic, $C$ can be calculated from cloud-free profiles. Its values are then compared to the frost index. As could be expected, $\frac{1}{C}$ diminishes with $\gamma$: that is, the signal is dampened when the window is covered with frost. An empirical fit of $\frac{1}{C}$ as a function of $\gamma$ can then be established (Mariage, 2015). This allows us to deduce the value of $C$ for each profile from the value of $\gamma$. The fitting coefficients were determined independently for each buoy when possible, since the frost index depends on $P_0$, which is buoy specific.

It should be noted however that when the frost is too thick ($\gamma \leq 0.05$), no usable signal is recoverable. This means that there were no exploitable lidar profiles in late December to early March. Furthermore, this frost correction method naturally causes uncertainty on the obtained value of $C$. Around $11\%$ of profiles have values of $\gamma$ between 0.1 and 0.3. In this case, Mariage (2015) estimates that the window frost correction leads to a $30\%$ error on $C$. A further $3\%$ of profiles have $0.05 \leq \gamma < 0.1$, in

which case the error on $C$ can be up to $60\%$. For $\gamma \geq 0.3$, the $C$ error tends towards the frost-free system constant determination error, which is around $10\%$ (Mariage, 2015). The system constant is used in the calculation of the attenuated scattering ratio, from which all cloud quantities are derived (Sect. 3.2). However, it is difficult to quantify the impact of its error on cloud detection, in part because it depends on the sign of the error. An overestimated $C$ would lead to under-detection of cloud layers, and vice versa. In practice, visual inspection of the profiles indicates that the cloud detection algorithm outlined below is robust to the errors that may be incurred through the window frost correction.

### 3.1.2  Receiver saturation due to reflective low clouds

The detectors used in the IAOOS lidar are avalanche photodiodes, and can reach saturation. This means that if they are exposed to a signal which is too intense, the photon count goes down. If the saturation is very intense, the photon count can even reach zero (Exc, 2018). Following saturation, the photon number count then slowly increases back up to its normal background value. Saturation is not usually an issue in most lidar operation situations; however during the Arctic summer, background noise levels are high due to shortwave radiation and the reflective sea ice and the signal reflected by the very low cloud cover is often enough to saturate the detector. This problem was observed from the very first deployment of the IAOOS buoys (Mariage, 2015). It translates visually into a lidar signal which dips below background noise levels at a certain altitude, and then slowly increases back to the background. Over the whole IAOOS period, approximately $30\%$ of profiles were concerned by this phenomenon.

A saturated profile may contain some geophysical data above the saturation altitude; therefore, it was important to correct this effect. We hypothesised that the saturated signal $S_{sat}$ resulted from the convolution of the "true" signal $S$ with a saturation impulse response function ($IRF$):

$$S_{sat}(z) = S(z) * IRF(z)$$

The goal was therefore to deduce $S$ from the measured profile, i.e. $S_{sat}$. A deconvolution algorithm was therefore put into place (Richardson, 1972; Refaat et al., 2008). The deconvolution process recovered useful signal from the saturated profiles in about a third of cases. In the remaining two-thirds, the "true" signal was only background noise. This represented an appreciable gain in data for the IAOOS campaign.

## 3.2  Derivation of cloud characteristics from raw lidar data

The lidar profile treatment program is a simplified version of the CALIPSO treatment algorithm described by Winker et al. (2009).

### 3.2.1  Attenuated scattering ratio calculation

The first step involves calculating the attenuated scattering ratio:

$$SR_{att} = \frac{(S-B) \cdot z^2}{C \cdot O(z) \cdot \beta_m(z) T_m(z)^2} = (1 + \frac{\beta_p(z)}{\beta_m(z)}) \cdot T_p(z)^2 \tag{1}$$

where

- $S$ is the raw signal;

- $B$ the background noise (calculated as the mean of the raw signal above 20 km, where there is no geophysical signal due to attenuation);

- $z$ the altitude above the lidar, which is at sea level;

- $C$ is the system constant, which varies with the lidar window frost as described above;

- $O(z)$ is the overlap factor between the lidar source and receiver: this factor is determined for each buoy as the average ratio of the raw signal to the calculated Rayleigh signal for very clear, cloudless days. The overlap creates a minimum height underneath which the signal cannot be resolved: a sort of lidar "blind zone";

- $\beta_p(z)$ and $\beta_m(z)$ are the particulate and molecular backscatter ratios at altitude $z$, respectively;

- $T_p$ and $T_m$ are the particulate and molecular transmission at altitude $z$, respectively.

The Rayleigh (molecular) backscatter and transmission are calculated according to Bucholtz (1995), using vertical temperature and pressure profiles from ERA5 reanalyses.

### 3.2.2 Cloud detection

Clouds are then detected by applying a threshold to $SR_{att}$, since in the absence of particulate attenuation the attenuated scattering ratio will be equal to 1 ($\beta_p = 0$, $T_p^2 = 1$). The initial threshold, $S_t$, is set to 1.1 at $z = 0$ and increases with altitude in order to take into account that noise increases on the vertical (Winker and Vaughan, 1994).

The base of a feature is detected when seven consecutive points are above the threshold. The top is detected either when $SR_{att}$ has fallen beneath the threshold and has stopped decreasing (a condition inspired by Winker and Vaughan (1994)) or when the signal is below the noise level. The noise level is defined as $2\sigma z^2$, where $\sigma$ is the standard deviation of the raw signal above 20 km. Assuming gaussian noise, 95% of pure noise fluctuations are therefore beneath this level.

Above the features, $SR_{att}$ will again be constant but equal to $T_f^2(z_{top})$, where $z_{top}$ is the top altitude of the features and $T_f$ its transmission, because of the particle attenuation. This means that new features above this feature will be missed unless the threshold is modified to take the feature attenuation into account. Therefore, above a feature, the threshold is updated to $T_f^2 \cdot S_t$.

Once detected, a feature is determined to be a cloud if its spread, defined as the ratio of maximum feature $SR_{att}$ to average below-feature $SR_{att}$, is greater than 100 (or 20 for higher-altitude layers for which average below-feature $SR_{att}$ is strongly impacted by noise).

### 3.2.3 Calculation of optical depth and lidar ratio

When the lidar beam goes through the cloud layer and reaches the particle-free air on the other side, the cloud transmission can be directly calculated as the ratio of the mean $SR_{att}$ above and below the cloud layer over a minimum of 20 points (or 300 m).

However, this was rarely the case during IAOOS, especially in the summer when the noise level is high. Over the whole IAOOS campaign, only $14\%$ of all features were transparent to the lidar. In all other cases, the cloud transmission $T_c^2$ was calculated from the integrated attenuated backscatter ($IAB$), assuming a constant lidar - or backscatter-to-extinction - ratio $S_c$ within the cloud layer:

$$IAB = \int\limits_{z_0}^{z_1} \beta_p(z) \cdot e^{-2\int_{z_0}^{z} \eta\alpha_p(z')dz'} dz = \frac{1}{2\eta S_c}(1 - T_c^2) \tag{2}$$

with $z_0$ and $z_1$ the bottom and top of the cloud, $\alpha_p$ the particle extinction coefficient and $\eta$ the multiple scattering coefficient (Platt, 1973). The $IAB$ can then be calculated from the attenuated scattering ratio and molecular backscatter (Winker et al., 2009) :

$$
\begin{aligned}
IAB \approx & \int\limits_{z_0}^{z_1} SR_{att}(z) \cdot \beta_m(z)dz \\
& - \frac{1}{2}(z_1 - z_0) \cdot (\beta_m(z_0)SR_{att}(z_0) + \beta_m(z_1)SR_{att}(z_1))
\end{aligned}
\tag{3}
$$

The (relatively few) cases where the cloud layer transmission could be independently calculated were used to derive values of the multiple-scattering lidar ratio $S^* = \eta S_c$ by inverting Eq. (2).

For both Rayleigh- and IAB-derived $T_c$, the cloud optical depth $\tau_c$ can then be deduced:

$$T_c = e^{-\eta \cdot \tau_c} \tag{4}$$

The multiple-scattering coefficient $\eta$ was assumed constant and equal to $0.8$, based on previous analyses of the IAOOS data (Mariage et al., 2017; Di Biagio et al., 2018).

### 3.2.4 Uncertainty and limits of the method

Equation (2) implies that as $T_c^2 \to 0$, $IAB \to \frac{1}{2\eta S_c}$. This means that for optically thick clouds, a small error on the value of IAB or $S_c$ risks propagating to a large error on COD. The error is also asymmetrical: an overestimation of IAB or $S_c$ yields a much worse result on COD than an underestimation of these same quantities. In practice, if the lidar ratio of a cloud of true optical depth $1.5$ is underestimated by $10\%$, the measured optical depth will be $\approx 1.1$. On the other hand, if it is overestimated by the same amount, the measured optical depth will be $\approx 2.2$. In some cases, overestimation of lidar ratio or IAB can even lead to negative $T_c^2$ values, which is non-physical and doesn't allow for the calculation of optical depth. In practice, therefore, this method is appropriate mainly for optically thinner cloud layers. We will refer to "low-IAB" cloud layers, for which the method does not lead to non-physical results (i.e., the cloud layer is thin enough that this method works well). This accounts for $42\%$ of all features. We will call "high-IAB" cloud layers those for which calculated $T_c^2$ is negative. These mathematically correspond to clouds with higher IAB, and therefore higher COD, than low-IAB cases. The inclusion of these high-IAB COD values in the statistic will be discussed in Sect. 4.3.

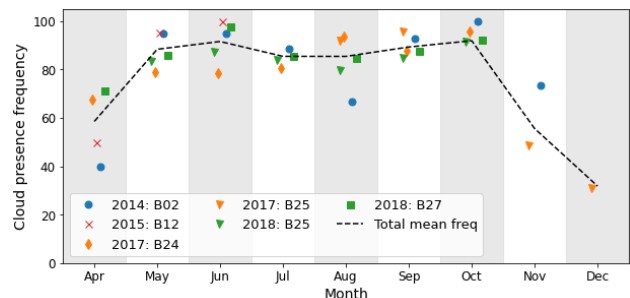

**Figure 2.** Monthly variation of low cloud frequency, defined as the number of profiles that contain at least one cloud layer with base lower than 2 km divided by the total number of profiles for the month, for five IAOOS buoys. The dashed line represents the total monthly cloud frequency over all IAOOS profiles. It is only calculated for months with more than 30 profiles in total.

Although uncertain in other respects, this COD calculation method has the advantage of being only faintly impacted by background noise levels. On the other hand, noise levels can have a strong impact on the cloud top determination. Tests with simulated lidar signals indicate that cloud top determination error reaches up to 150 m for typical summer noise levels and optically thicker clouds ($\tau_c \approx 2.5$). This error is much lower for low noise levels, such as are found in the high Arctic during the polar night (October - March). This difference must be kept in mind when interpreting seasonal variation of cloud geometrical thickness (Sect. 4.2).

## 4   Seasonal variability of Arctic low cloud properties during IAOOS

### 4.1   Frequency of cloud presence

IAOOS data confirms that low clouds (i.e., with a base under 2 km) are very frequent in the Arctic, especially in the summer. Average monthly cloud frequency from March to December, defined as the average of monthly ratios of profiles containing at least one cloud with base lower than 2 km to all profiles, is 75%. This value is coherent with previous statistics of cloud fraction above 80°N derived from satellites, for example Wang and Key (2004) and Curry et al. (1996), which usually give a global annual cloud cover of around $60 - 70\%$, with a maximum in summer and a minimum in November - April.

Observed seasonal variation of cloud fraction can differ strongly between satellites (Wang and Key, 2004; Zygmuntowska et al., 2012). Chan and Comiso (2013) found large disagreements between MODIS and CALIOP in the Arctic, for example, especially over sea-ice and during the polar night. This is because MODIS finds it difficult to differentiate between the surface and the clouds when relying only on IR channels. On the other hand, Blanchard et al. (2014) shows that there is good general agreement and similar trends in cloud fraction over Eureka (Nunavut, Canada) between CALIOP, MODIS, CloudSat and the IIR instrument aboard CALIPSO, with a global maximum in September - November and a minimum in March - May. However, discrepancies between passive and active instruments remain (Blanchard et al., 2014). Ground-based measurements play a key part in quantifying seasonal cloud cover variability in the Arctic, although they are often sensitive primarily to

lower-level clouds. Averaging visual observations from ships and ice-camps above 80°N, Hahn et al. (1995) found that cloud cover was globally stable around 60% in winter, increasing to 80% from April to June, and decreasing again from September to November. A maximum of 85% was reached in August/September. The combined lidar-radar measurements at SHEBA give slightly higher values of 70% in winter and 90% in summer, with an earlier transition (February to April) and a peak in September (Intrieri et al., 2002b).

The results of IAOOS dataset are shown in Table 3 and Fig. 2. Note here that the number of profiles available for each month is variable, both because of the more favorable operation conditions in the summer and the timing of the buoy deployment (usually in May). As such, there are more than 200 profiles from May to September, around 100 in April and October, and less than 54 in November and December (months with less than 30 profiles, i.e. January, February, and March, are not treated in this article). Care must therefore be taken in analysing the results of late autumn and winter. A 90% confidence interval for the cloud occurrence frequency can be estimated from a Bayesian calculation, assuming that the number of cloudy profiles follows a binomial distribution and supposing an appropriate a priori distribution for the cloud frequency from the literature (Appendix A).

The IAOOS data shows a similar trend as the literature, with generally higher cloud cover values. From May to October, clouds are present over 85% of the time (Fig. 2). In contrast to the previous ground-based climatologies outlined above, there are two peaks at more than 0.9 in the monthly cloud frequency, although they differ little from the summer baseline. The first is in June, which has a mean cloud frequency of 0.92 and a confidence interval of (0.88 − 0.94). The second peak is in October, also with a mean cloud frequency of 0.92 but with a slightly wider confidence interval (0.85 − 0.95) because of the lower number of profiles. This is reminiscent of the results of Zygmuntowska et al. (2012), from CALIPSO data, which show a peak in cloud occurrence above 0.9 in October. July and August have slightly lower cloud frequency values (0.85 (0.82 − 0.88) and 0.85 (0.8 − 0.89) respectively). However, since there is non negligible overlap between the confidence intervals of June/October and the other summer months, it is difficult to draw solid conclusions as to May - October variability.

In the IAOOS dataset, April and November appear to mark a sharp transition in cloud occurrence frequency from the summer values. April has a cloud frequency of 0.59 (0.52 − 0.67) while the cloud frequency in November is 0.56 (0.48 − 0.68). While the confidence intervals are quite wide here due to the lower number of profiles, there is no overlap with the summer confidence intervals. This suggests that the lower cloud frequencies observed during the months of April and November is meaningfully different from that of the months of May through October. December cloud frequency is lower still, at 0.32 (0.29 − 0.51). Note however the width of the confidence interval and the fact that the December data corresponds to a single year of measurement (2017).

It is not possible to robustly quantify interannual variability in Arctic cloud cover from the IAOOS dataset since there are at most four years of data for each month. Qualitatively, however, the April - May transition in cloud frequency observed by the buoys is quite variable. In 2014, the B02 buoy observed a very sharp spring transition in cloud frequency: from 0.4 (0.35 − 0.6) in April 2014 to more than 0.9 (0.89 − 0.97) in May and June 2014 (blue circles, Fig. 2). On the other hand, this transition was much more gradual in 2017 (buoy B24, orange diamonds). The June 2017 cloud frequency is less than 0.8 (0.69 − 0.85),

| Month | $N_p$ (#) | $\frac{N_{ml}}{N_{profiles}}$ (%) | First cloud base (%) | | | |
|---|---|---|---|---|---|---|
| | | | $<$ 120m | 120− 500m | 500m −2km | 2− 5km |
| Apr | 94 | 4 | 96 | 0 | 2 | 2 |
| May | 359 | 4 | 95 | 2 | 1 | 1 |
| Jun | 330 | 8 | 87 | 8 | 3 | 1 |
| Jul | 342 | 14 | 93 | 1 | 3 | 3 |
| Aug | 205 | 12 | 91 | 2 | 5 | 2 |
| Sep | 251 | 10 | 90 | 5 | 4 | 1 |
| Oct | 98 | 13 | 98 | 2 | 0 | 0 |
| Nov | 54 | 2 | 93 | 3 | 3 | 0 |
| Dec | 44 | 0 | 93 | 7 | 0 | 0 |

**Table 2.** Cloud multiple layer and base characteristics for all profiles from April to December. $N_p$ is the total number of lidar profiles for each month (for all years and buoys), and $N_{ml}$ is the number of profiles containing multilayered clouds. The last four columns represent the % of first layer cloud bases in each altitude range. The 120 m cutoff corresponds to the minimum altitude at which the lidar overlap factor can be corrected for all buoys. Cloud bases above 5 km, which correspond to "high-level" clouds in many reanalyses such as ERA5, are not included because the lidar range in perfectly clear daytime conditions is only 4.4 km (Sect. 2.1).

overlapping significantly with the May 2017 cloud frequency confidence interval of $(0.56-0.78)$. This is not an effect of spatial variability as both B02 and B24 were drifting in the Atlantic sector of the Arctic (Fig. 1).

It has been observed from satellite data that the Atlantic sector is the cloudiest part of the Arctic Ocean (Liu et al., 2012; Wang and Key, 2004). This is linked to the low pressure systems and the storm tracks arriving from the northern Atlantic Ocean. Since most of the IAOOS buoys drifted in this sector, the IAOOS dataset must be regarded as most representative of these specific conditions, and not of the ocean-wide cloud characteristics.

Furthermore, the results above pertain to the low cloud cover, i.e. clouds with a base underneath 2 km. Clouds with a base between $2-5$ km are much rarer in the IAOOS dataset, occurring only $3\%$ of the time from March to December, with a peak at $8\%$ in July. However, as the lidar signal is often dampened by the first cloud layers, IAOOS statistics of cloud cover above 2 km are expected to be biased low.

## 4.2 Cloud geometrical properties

Multi-layer clouds were detected $7\%$ of the time by the IAOOS lidar over the course of the campaign. This value is small compared to previous observations: for example, Liu et al. (2012) find that multi-layer clouds are present $20\%$ of the time year-round, with very low seasonal variation. These results are drawn from satellite observations and Liu et al. (2012) note that they are also underestimated. Ground-based measurements generally attest to frequent multilayering in the summertime,

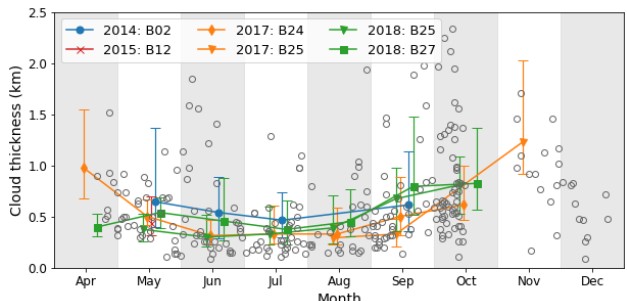

**Figure 3.** Monthly evolution of first layer cloud geometrical thickness (in km), for five IAOOS buoys. The markers represent the median value, and the whiskers indicate the 25th and 75th percentiles. The open circles represent individual cloud thickness values where the lidar signal sees through the cloud layer, i.e. the cloud top is clearly detected. The median, 25th and 75th percentiles are only calculated when more than 15 data points are available.

with layers separated by several hundred meters (Curry et al., 1988, 1996). SHEBA measurements even show that multi-layer clouds exceeded single-layer clouds in June and July 1998, and occurred on average $45\%$ of the time over the whole experiment period (Intrieri et al., 2002b). IAOOS measurements also attest to a higher frequency of multiple layered clouds in summer: they occur more than $10\%$ of the time July - October, and only $4\%$ of the time in April and May (Table 2). Only one IAOOS profile contains multilayered clouds in November, and none in December. Despite the low number of total profiles in these months,

these values are different from the July multilayered cloud frequency at a statistically significant level: for November, Fisher's exact test yields a p-value of $0.007$ (Fisher, 1922). IAOOS measurements strongly underestimate frequency of multilayered clouds due to the fact that the lowest cloud layer entirely attenuates the lidar signal in most profiles. Furthermore, cloud layers separated by less than 300 m were counted as one in the IAOOS data treatment in order to have a better estimation of cloud transmission (Sect. 3.2.3). However, the robust measurement of the geometry of the first cloud layer derived from IAOOS

measurements base is a useful statistic. Indeed, the base of the lowest cloud layer is expected to have the strongest impact on surface radiative fluxes as compared to higher cloud layers. Hereafter, all cloud statistics refer to single cloud layers; in most cases, the lowest.

    Clouds in the IAOOS dataset are extremely low, with little seasonal variability. From April to December, at least $85\%$ of first layer clouds have a base below 120 m, which is the minimum altitude at which the lidar overlap factor can be corrected for all

buoys (Table 2). The median base altitude is therefore at 120 m in nearly every month. During ASCOS, which took place in August 2008, the lowest cloud base distribution peaked beneath 100 m (Tjernstrom et al., 2012). Median first cloud base from SHEBA measurements (Shupe et al., 2007) was also less than 120 m for all months except March (179 m) and April (209 m). Nevertheless, higher-altitude first cloud layers were more frequent than during IAOOS, especially in spring to early summer (Intrieri et al., 2002b).

On the other hand, Fig. 3 highlights a significant difference in measurements of single-layer cloud geometrical thickness between summer (May to September) and the months of April, October and November. The median cloud thickness from

June to August ranges between 360 and 390 m, whereas it is nearly 750 m in October and March, and more than 1 km in November. This difference appears significant at a statistical level. The Mann-Whitney $U$ for the July and October cloud thickness distributions was 9834.5 (with sample sizes $n_1 = 355$ and $n_2 = 104$), yielding a p-value $< 0.001$ (Mann and Whitney, 1947). The same is true for July and April ($U = 5940.5$, $n_1 = 355$ and $n_2 = 60$, p-value $< 0.001$).

As explained in Sect. 3.2.3, it is expected that summer cloud thickness would be underestimated by up to 150 m due to higher noise levels in this period. However, this is too small an error to explain the different median values observed between summer and spring/autumn. Furthermore, these values and trends are coherent with previous studies of single-layer clouds at Barrow and Eureka. For example, the average thickness of single-layer clouds at Barrow from June to August 2000 was 320 m while the September average was 550 m (Dong and Mace, 2003). Over the 2005 to 2008 period the average single-layer mixed-phase cloud thickness at Eureka varied from 200 m to 700 m with maxima in autumn and minima in spring (de Boer et al., 2009). Total thickness of all clouds, single-layered or not, may however be much larger. During SHEBA, median total cloud thickness from radar data was above 1 km in every month, with peaks at around 3 km in April and October (Shupe et al., 2007). These values are from 3 (March/April) to 7 (July/August) times larger than the IAOOS monthly median values.

## 4.3 Cloud optical properties

As noted in Sect. 3.2.3, cloud layers for which both IAB and $T_c^2$ are determined independently can be used to calculate the multiple-scattering lidar ratio $S^*$. In total, there were 207 such cloud layers during the IAOOS period, covering the March to December period. They are shown in Fig. 4a, along with the median and the 25th and 75th percentiles for each month. The global median is 17.5 sr, with $90\%$ of values falling in the $7 - 38$ sr range. Although the spread is quite large, these results are consistent with cloud lidar ratio values found in the literature. For example O'Connor et al. (2004) found that $S^*$ values ranged between 14.5 and 16.5 sr for low water clouds; for ice or mixed-phase clouds, the range was $5 - 40$ sr, very similar to IAOOS results.

The seasonal variation of $S^*$ is statistically significant: the median $S^*$ for the summer months (JJA) was 23 sr versus 15.5 sr in the autumn (SON). The Mann-Whitney $U$ is 4953.5, with $n_1 = 67$, $n_2 = 98$, yielding a p-value of $< 0.001$ (Mann and Whitney, 1947). There are two possible causes for the observed variability in $S^* = \eta S_c$: changes in the multiple scattering coefficient $\eta$ or $S_c$. $\eta$ decreases with cloud temperature (Garnier et al., 2015) while $S_c$ depends on cloud microphysical properties, among which cloud droplet effective radius and phase. In the absence of additional measurements, it is difficult to determine which one has the largest impact here, as well as the ultimate physical cause of variation. The monthly median values were then used to calculate COD (Sect. 3.2.3).

The average single-layer COD during IAOOS excluding high-IAB cases was 0.9, with values ranging from 0.3 to 2.1. These values are small when compared to previous satellite and ground based studies in the Arctic. But as noted in Sect. 3.2.4, the retrieval method used for calculating COD from the IAOOS lidar data when the signal is fully attenuated is not suited to optically thick clouds: the rough upper bound of COD which can be measured through this method is 2. As almost $20\%$ of cloud layers observed during the campaign were high-IAB, this likely has a non-negligible impact on results. Furthermore, contrarily to satellite data, IAOOS values are single-layer, not whole column, COD. The contribution of the first layer to total column

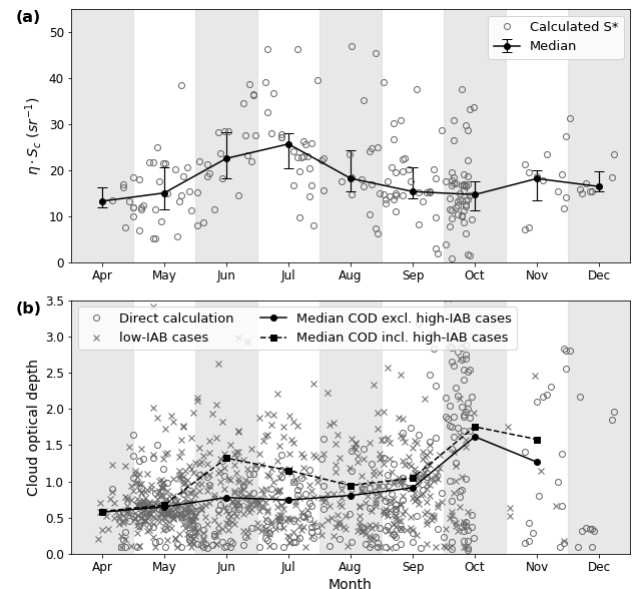

**Figure 4.** Panel a: monthly variations of lidar ratio values over the IAOOS campaigns. The open circles represent the measurements. The filled markers represent the monthly medians, with the whiskers indicating the 25th and 75th percentiles. Panel b: monthly evolution of single-layer COD, for five IAOOS buoys. Open circles represent the Rayleigh-derived cloud optical depths. Crosses correspond to the low-IAB COD values (Sect. 3.2.4) Filled markers represent the monthly medians, when high-IAB cases are excluded (circles) or included (squares). These medians are calculated when more than 15 data points are available.

COD is discussed in Sect. 5.3. It is therefore understandable that previous studies gave larger COD values. For example, Curry et al. (1996) cites a range of $2 - 24$ with an average of $8$ in summer. Wang and Key (2004) also finds that monthly mean COD (from 1982 - 1999) varied from $4$ to $6$ in the AVHRR data over the Arctic Ocean. From ground-based lidar measurements at SHEBA, Turner (2005) shows that $63\%$ of clouds were single-layer with an optical depth $< 6$, and that optically thin clouds
tended to be predominantly composed of ice.

Single-layer COD appears to vary seasonally (Fig. 4b). Excluding high-IAB cases, the monthly median COD appears to be almost constant from April to September, and largest in October - November (filled circles). However, this is in part because of the low noise levels in these months as compared to the summer. In October - December, i.e. the months with no sunlight, more than $50\%$ of cloud layers were transparent to the lidar. This proportion is less than $10\%$ in May to July. The COD can therefore
be directly calculated for optically thick clouds from late September - December but not in other months. This is visible in Fig. 4b: in late September/October, there is a sudden apparition of directly-calculated COD values (open circles) greater than 2. The IAB method, which is an alternative to the direct method of calculating COD when the signal is fully attenuated by the cloud, is mainly suited to optically thin clouds (Fig. 4, grey crosses). This creates bias between summer months, for which the COD calculation is limited by noise levels to optically thin clouds, and October - December, during which higher COD values
can be calculated.

| Month | Number of profiles | Cloud fraction (%) | Median temperature (°C) | | |
|---|---|---|---|---|---|
| | | | Cloudy | Cloudless | Δ |
| Apr | 94 | 59 | -17.7 | -21.2 | 3.5 |
| May | 359 | 88 | -9.9 | -13.6 | 3.7 |
| Jun | 330 | 92 | -1.5 | -1.5 | 0 |
| Jul | 342 | 85 | -0.1 | -0.5 | 0.4 |
| Aug | 205 | 85 | -0.9 | -1.1 | 0.2 |
| Sep | 251 | 89 | - 3.7 | -6 | 2.3 |
| Oct | 98 | 92 | -6.6 | -14.6 | 8. |
| Nov | 54 | 56 | -16.7 | -25 | 8.4 |
| Dec | 44 | 32 | -27.9 | -28.5 | 0.6 |

**Table 3.** Monthly median temperature for cloudy and cloudless profiles from April to December over the whole IAOOS period. Cloudy profiles contain at least one cloud with a base underneath 2 km. Cloudless profiles contain no clouds, or (very rarely) higher level clouds. Δ is the difference between cloudy and cloudless profile median temperatures.

To overcome this problem, the COD of high-IAB cloud layers was set to 2. This value was chosen as it is the 95th percentile of CODs calculated for low-IAB layers, and high-IAB cloud layers are as a group expected to have higher COD than low-IAB layers. The monthly median COD was then calculated including these high-IAB cases (Fig. 4, filled squares). This correction is not quantitatively robust as the value of 2 is arbitrarily chosen, not calculated. However, it accounts for the fact that high-IAB

cloud layers exist, and are expected to have higher COD than low-IAB cloud layers, in the calculation of the median. This is helpful for examining the seasonal trend, which otherwise is biased by the presence of noise.

It creates a significant difference in June and July, the months in which the percentage of high-IAB cloud layers is the highest. With this correction, the median monthly COD exhibits two peaks (June and October) and a minima in April. The October peak is however still the annual maximum, and does not appear to be strongly impacted by the inclusion of high-IAB

cloud layers. Previous satellite measurements have exhibited a pattern of higher COD in spring and autumn, for instance May and October for the AVHRR data (Wang and Key, 2004) over the Arctic Ocean. The IAOOS dataset exhibits this October peak in single-layer COD. Another peak in June appears possible, although the IAOOS measurements are very uncertain in this month.

## 5  Cloud impact on surface temperatures and radiative balance

### 5.1  Impact of clouds on surface temperatures during IAOOS

IAOOS lidar profiles can be split into two groups: "cloudy" profiles containing at least one low cloud with a base $< 2$ km and "cloudless" profiles which contain either no cloud, or higher level clouds. Note that less than 2% of all clouds had a base higher than 2 km (Sect. 4.2). The temperatures measured by the buoy meteorological station during each lidar profile acquisition can be compared to estimate the effect of the presence of low clouds on surface temperatures.

The 2 m temperature distributions of cloudy and cloudless profiles differ significantly in October-November and April (Table 3). The Mann-Whitney test p-value is less than $0.05$ ($< 0.001$ for November) and the common language effect size is more than $70\%$ ($> 80\%$ for October and November). For all of these these months, the 2 m temperature is much lower for cloudless than for cloudy profiles. Indeed, the difference between the medians is of 8°C for the autumn months and around $4 - 7$°C in the spring (Table 3). This difference is probably not due solely to radiative processes, as cloudy situations in the Arctic winter are also associated with the passage of storms, which bring warm, moist air with them. However, as seen in Sect. 4.3, IAOOS-measured CODs are larger in October/November than April. Since emissivity increases with optical depth, this supports a larger surface warming in autumn than in spring. The months with the lowest median temperature difference between cloudy and cloudless profiles are June, July and August. In fact, the temperature distributions are statistically indistinguishable in these months from the relatively few measurements we have access to here. In particular, there is no month in which cloudless profiles are warmer than cloudy profiles, even though clouds are known to exert negative radiative forcing from late June to early July.

As noted before, clouds are naturally not the only factor impacting surface temperatures or even the downwards longwave radiative flux. Large-scale circulation is also important: for example, high geopotential at 200 hPa is linked to a warming of the troposphere through subsidence, which increases the longwave radiative flux received at the surface (Ding et al., 2017). It is therefore important to check that cloudy and cloudless lidar profiles do not sample different surface pressures. The IAOOS buoys were equipped with barometers as well as temperature sensors. It appears that surface pressures for cloudy and cloudless profiles are not different at a statistically significant level, with the exception of August and November. In both of these months, the lidar profiles that contain clouds appear to coincide with markedly higher surface pressures than those that don't contain clouds ($+12$ hPa, Mann-Whitney test p-values $< 0.005$). As surface temperatures in the two groups differ strongly in November but not in August, however, surface pressure does not appear to be a confounding factor for surface temperature and cloud occurrence.

In the following sections, we look at the summer surface radiative balance in order to gain a better understanding of the mechanisms behind this seasonal variation in temperature difference between cloudy and cloudless profiles. First, the link between the net surface longwave flux and the presence of clouds is investigated (Sect. 5.2.1) from compared N-ICE and IAOOS measurements. Then, the influence of other factors such as solar zenith angle, temperature and COD on downwards shortwave and longwave fluxes during the N-ICE2015 April to June period is explored (Sect. 5.3). Lastly, the discussion of the

net cloud radiative forcing at the surface is extended to the months of July and August using a simple parametrisation (Sect. 5.4).

## 5.2 Influence of the presence of clouds on the surface net longwave radiative flux

### 5.2.1 Identification of two summer longwave radiative modes from IAOOS and N-ICE data

The 2 m temperature difference between cloudy and cloudless autumn/winter profiles exposed in Sect. 5.1 is consistent with previous studies. Indeed, it is now well attested that the Arctic climate exhibits two distinct states during the winter, which are distinguished through the surface net longwave flux (netLW) values. The bimodality of netLW was first observed during the SHEBA measurement campaign over the January-February 1998 period (Stramler et al., 2011) and has since been confirmed

Arctic-wide by satellite observations (Cesana et al., 2012). The "radiatively clear" mode (netLW $< -30$ W m$^{-2}$) is associated with strong radiative cooling, high pressures and low temperatures. Clouds may be present but are optically thin and mainly composed of ice. The "opaquely cloudy" mode is characterised by low pressures and relatively higher temperatures, and often associated with so-called "moisture and temperature intrusions" from the midlatitudes (Woods et al., 2013). Clouds are then liquid or mixed-phase, and optically thick. These intrusions are one of the main drivers of interannual variability of netLW,

with a contribution of about $40\%$ (Woods et al., 2013).

Here, we used radiative flux data from the N-ICE field campaign (second period, April - June 2015) to complement the IAOOS lidar observations (Hudson et al., 2016). Measurements from the first period (January to March 2015) of N-ICE have already been shown to confirm the wintertime bimodality of the netLW distribution (Graham et al., 2017). This result is replicated in Fig. 5b. A more striking point is that the netLW distribution is also bimodal in spring to early summer (Fig. 5c).

During this period, netLW values range from $-90$ to $0$ W m$^{-2}$. The most predominant netLW mode, containing around $80\%$ of data points, is centered around $-11$ W m$^{-2}$, while the other is centered around $-72$ W m$^{-2}$. As a IAOOS buoy drifted near the main ice camp during April-June 2015, the IAOOS profiles were used to determine whether the sky was cloudless or cloudy at a given moment. The comparison with netLW measurements is represented in Fig. 5a. Low netLW values ($< -60$ W m$^{-2}$) are associated with IAOOS profiles that are cloudless at least up to $\approx 5$ km, which is the maximum range of the lidar.

Meanwhile, profiles containing at least one low level cloud (grey lines) corresponded to netLW values larger than $-20$ W m$^{-2}$.

This shows that the observed low netLW mode corresponds to a cloudless state and the high netLW mode to a cloudy state. By analogy with the previously established winter radiative states, we name the spring/summer low-netLW mode "radiatively clear" and the high-netLW mode "opaquely cloudy". However, these two modes differ from their winter analogues in several ways. Firstly, the netLW mode values are lower than in the winter. Indeed, both the downwards and upwards components of

455 the longwave flux (LWd and LWu) increase from winter to summer. However, LWu increases more than LWd in both modes, causing a shift to lower netLW values. Secondly, the opaquely cloudy mode is much more frequent in spring/summer than in the winter, representing a large majority of cases. This is coherent with the fact that cloud frequency is much higher in spring/summer than in winter, with a transition in April (Sect. 4.1). Thirdly, the difference between the two states is $\approx 60$

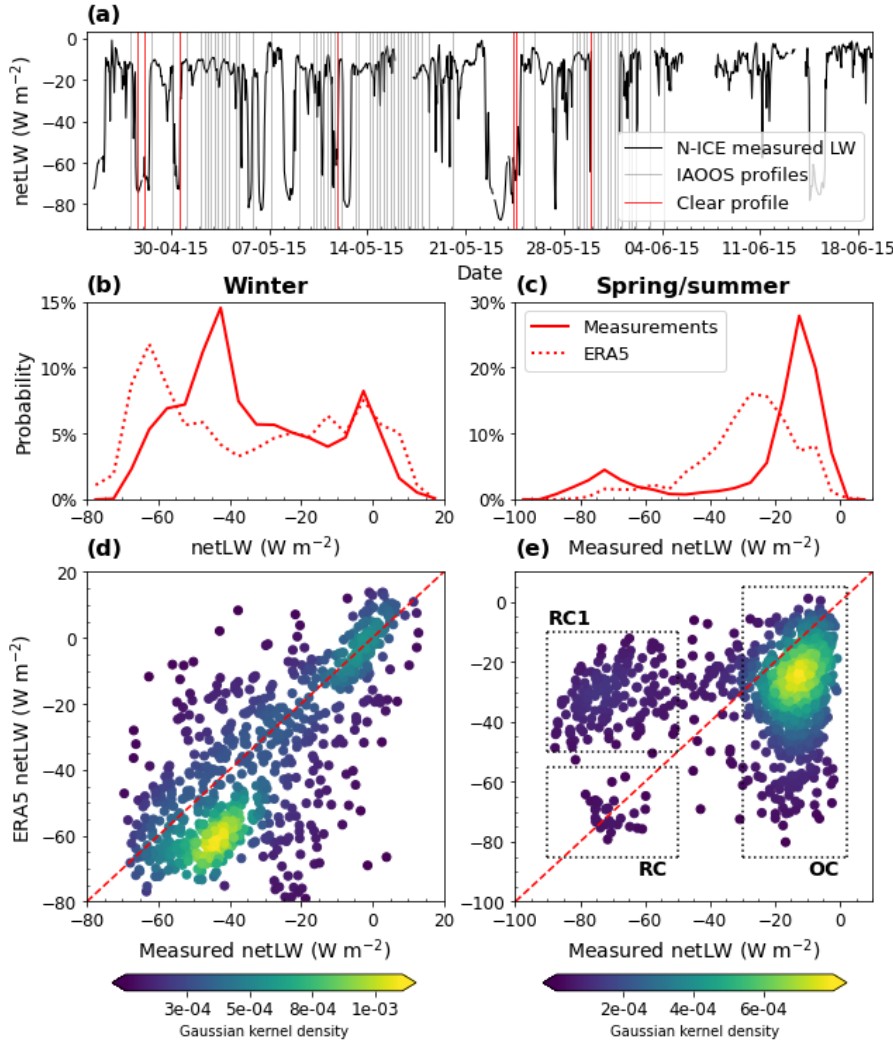

**Figure 5.** Panel a: time series of surface net longwave measurements during the N-ICE field experiment (second period, April-June 2015). The vertical lines indicate the time of IAOOS lidar profiles, with red lines corresponding to cloudless profiles. Panels b and c: histogram of the measured (filled line) and ERA5 (dashed line) net longwave flux during the N-ICE winter (b) and spring/summer (c) campaign periods. Panels d and e: hourly ERA5 vs measured net longwave in during the N-ICE winter (d) and spring/summer (e) campaign periods, with red dashed line indicating the 1:1 line. The colour corresponds to point density as calculated by a Gaussian kernel. For panel (e), three zones have been outlined. Zone "OC" contains points belonging to the opaquely cloudy mode of the measured netLW distribution. Zones "RC1" and "RC2" contain points belonging the radiatively clear mode of the distribution in April and May (RC1) and June (RC2).

W m$^{-2}$, much larger than in the winter. This implies that clouds have a larger longwave warming effect in the spring/summer than in the winter, probably linked to larger liquid contents and higher cloud temperatures in this season.

### 5.2.2 Representation of the two modes in the ERA5 reanalyses

The two atmospheric winter states (radiatively clear and opaquely cloudy) are not well reproduced by models (Cesana et al., 2012; Pithan and Mauritsen, 2014; Graham et al., 2017). In fact, it has been suggested that representing the bimodality of the netLW, pressure and temperature distributions in the wintertime is a key quality criterion for models. ERA-Interim and its suc-

465 cessor, ERA5, are among those that partially achieve this (Graham et al., 2017). This is visible in Fig. 5d. The opaquely cloudy state lies on the 1:1 line and is therefore well represented. However, the radiatively clear netLW values are underestimated by about $15$ W m$^{-2}$. This is mainly due to an error in the upwards component of the longwave flux. Indeed, ERA5 overestimates the clear mode 2 m temperature by about $5$ K; its measured value is $T_{meas} = -32°$C (Graham et al., 2017), while the ERA5 clear mode temperature is $T_{ERA5} = -27°$C. This leads to an error on the longwave upwards flux at the surface (LWu) of:

$$\Delta(\mathrm{LWu}) = 4\epsilon\sigma \cdot (T_{ERA5} - T_{meas}) \cdot (T_{meas} + 273.15)^3$$

$$\approx 15.6 \text{ W m}^{-2} \tag{5}$$

with $\epsilon$ the surface emissivity, which is assumed to be $0.99$ (Walden et al., 2017). The result of Eq. (5) is in line with the observed netLW error. It should be noted that this overestimation of near-surface temperatures in clear, stable winter conditions, leading to an underestimation of netLW, is a feature shared by the six reanalyses evaluated by Graham et al. (2019) using the N-ICE campaign data.

In the spring/summer period, Graham et al. (2019) further notes that ERA5 is the least biased of the six evaluated reanalyses with regards to netLW, but has the worst correlation coefficient ($R = 0.15$). Indeed, we find that ERA5 fails to represent the two spring/summer netLW modes. The ERA5 netLW distribution is not bimodal (Fig. 5c) and does not align with the measurements (Fig. 5e). Three zones have been outlined on figure 5e to aid with the following discussion of ERA5 spring/summer netLW error. Zone OC corresponds to measured opaquely cloudy values over all spring/summer. The opaquely cloudy mode

is somewhat reproduced by ERA5 (yellow dots denoting a peak in the calculated gaussian kernel density), although its values are underestimated by $11$ W m$^{-2}$ on average. The two other boxes correspond to measured radiatively clear values from April/May (RC1) and June (RC2) respectively. June values are well reproduced by ERA5. However, ERA5 vastly overestimates radiatively clear netLW in April and May: there is a $40$ W m$^{-2}$ difference with measurements in these month (Fig. 5e, RC1).

The difference in ERA5 netLW values between radiatively clear April/May (RC1) and June (RC2) points is due to the downwards component of the longwave flux (LWd). ERA5 LWd is fairly close to measured values in RC2, but is overestimated by $\approx 53$ W m$^{-2}$ in RC1. This is partly compensated by a $14$ W m$^{-2}$ error on LWu in April/May, similar to what is observed during the winter. Ultimately, the overestimation of LWd in RC1 is due to a faulty representation of cloud fraction in April/May. The ERA5 mean low cloud cover in RC1 is $0.96$, even though measurements indicate a radiatively clear, and therefore cloudless,

situation. On the other hand, mean low cloud cover in RC2 is $0.06$: ERA5 has correctly identified that the sky was cloudless.

In conclusion ERA5 overestimated low cloud cover in April and May, but not June, leading to the observed errors in netLW. More investigation is required as to the ultimate source of this error.

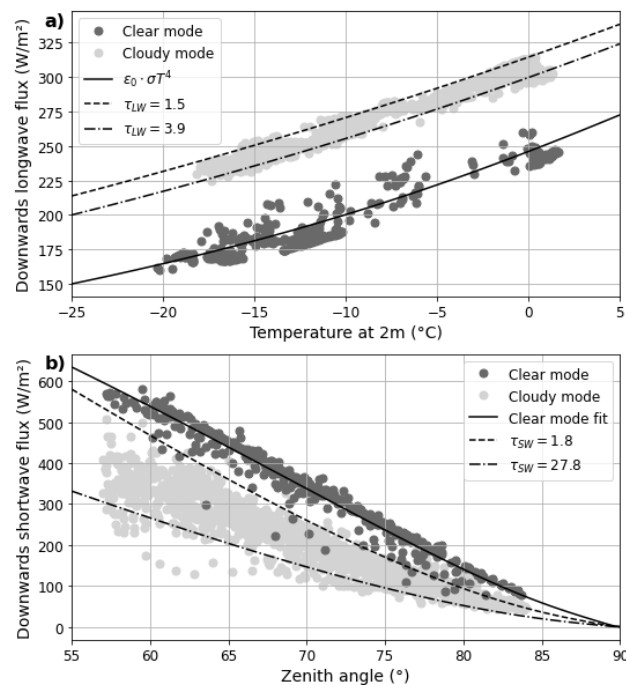

**Figure 6.** Panel a: longwave downwards radiative flux with near-surface (2 m) temperature as measured during the spring/summer period of the N-ICE field campaign. Dark grey points correspond to values for which $\mathrm{netLW} < -50$ W m$^{-2}$ ("radiatively clear" mode) while for light grey points $\mathrm{netLW} > -20$ W m$^{-2}$ ("opaquely cloudy" mode). The filled line correspond to the results of a simple parametrisation of LWd (Eq. (6)) in the absence of clouds, while the dashed lines represent the results of the parametrisation for $\tau_{LW} = 1.5$ and $\tau_{LW} = 4.1$. Panel b: same, for shortwave downwards radiative flux vs solar zenith angle. The dashed lines are the results of Eq. (7) for $\tau_{SW} = 1.7$ and $\tau_{SW} = 28.2$. For both panels, points are 30-minute averages of measurements.

## 5.3 Variability of cloud impacts on the downwards radiative fluxes during N-ICE2015

In the Arctic summer, clouds impact the surface radiative budget in two competing ways: they have a longwave warming
effect and a shortwave cooling effect. In Sect. 5.2.1, the N-ICE2015 April-June netLW distribution was shown to be bimodal, with the first mode corresponding to the presence of clouds in the IAOOS profiles and the second to their absence. However, other factors than the absence or presence of clouds may impact the surface radiative fluxes, both shortwave and longwave. In this section, the influence of variables such as the solar zenith angle, COD and surface temperature on the downwards fluxes (both longwave and shortwave) from the N-ICE2015 April-June period is explored and parametrisations of these fluxes are
introduced.

The longwave effect depends on cloud temperature and phase. Warm, liquid-containing clouds are optically thicker and have much more radiative impact than cold, ice-containing clouds (Shupe and Intrieri, 2003). This is most likely the reason behind the greater difference between netLW modes observed in the spring/summer ($\approx 60$ W m$^{-2}$) N-ICE measurement period as

compared to the winter ($\approx 40$ W m$^{-2}$). The shortwave radiative forcing depends on cloud characteristics as optically thick clouds have higher albedos. It also depends on the solar zenith angle $\theta$ and, to a lesser extent, the surface albedo $\alpha$, due to reflections between the bright surface and the clouds (Shupe and Intrieri, 2003).

As shown in Sect. 5.2.1, netLW values can be used to discriminate between "radiatively clear" and "opaquely cloudy" instants. The downwards longwave (LWd) and shortwave (SWd) flux components in these two modes are then compared in order to evaluate the impact of clouds on the surface. We will use the following simple estimates of LWd and SWd as a complement to the N-ICE flux measurements (Hudson et al., 2016).

- Schematically, the atmosphere can be seen as a cloud layer with emissivity $\epsilon_c$ overlying a cloudless atmospheric layer with emissivity $\epsilon_0$. If both layers are emitting at temperature $T_{2m}$, this yields the following expression for LWd:

$$\text{LWd} = [\epsilon_0 + \epsilon_c(1 - \epsilon_0)] \cdot \sigma \cdot T_{2m}^4 \tag{6}$$

The cloud emissivity can simply be expressed as $\epsilon_c = 1 - e^{-\tau_{LW}}$ with $\tau_{LW}$ the longwave COD. Several simple parametrisations exist for $\epsilon_0$; here, we choose $\epsilon_0 = 0.83 - 0.18 \cdot 10^{-0.067 e_0}$, with $e_0$ the near surface water vapour pressure, which was fitted from summer data at Sodankylä, Finland (Niemelä et al., 2001a). This shows good correspondence to the N-ICE clear mode data (Fig. 6a). In fact, equation (6) corresponds to a model introduced by Schmetz et al. (1986) under two simplifying assumptions. First, that the cloud cover is equal to 1, which is reasonable in the cloudy mode. Second, that the cloud base and two-meter temperatures are approximately equal. This is justified by cross-comparison of the N-ICE (second period) radiosonde data with the IAOOS lidar profiles: the overwhelming majority of lowest layer clouds have a base beneath 120 m and the median difference between surface and 100 m temperature in the radiosonde profiles is only 1.3°C (with 90% of values falling in the range $0.6 - 2$°C).

- SWd can be calculated from the downwards shortwave flux in the absence of clouds F$_0$ and the cloud correction or cloud broadband transmittance factor $T_c$:

$$\text{SWd} = \text{F}_0(\theta) \cdot T_c(\theta, \tau_{SW}, \alpha) \tag{7}$$

$F_0$ depends on atmospheric gas and aerosol content and is usually parametrised to fit to local data (Reno et al., 2012; Kambezidis et al., 2017). Here, the fit to N-ICE clear mode data is shown on Fig. 6b (filled black line). $T_c$ has been modeled in numerous ways, the simplest depending solely on cloud cover (Niemelä et al., 2001b), while more complicated expressions have been derived from the output of radiative transfer models. Here we used the parametrisation of Fitzpatrick (Fitzpatrick et al., 2003), which assumes a cloud cover of 1 and depends on the solar zenith angle $\theta$, the surface albedo $\alpha$ and the shortwave COD $\tau_{SW}$. We chose to use a fixed value of $\alpha = 0.8$, as the measured albedo over the N-ICE second period varied from $0.75 - 0.84$ and the model performs poorly for albedos above 0.83 (Fitzpatrick et al., 2003).

Downwards longwave radiative flux increased with near-surface temperature $T_{2m}$ and downwards shortwave flux decreased with $\theta$ in both radiatively clear and opaquely cloudy modes during the N-ICE April-June measurement period (Fig. 6). This evolution is well reproduced by Eqs. 6 and 7. Furthermore, there is a marked difference in downwards flux between points identified as radiatively clear and opaquely cloudy for both the longwave and shortwave components. In accordance with a cloud longwave warming effect, radiatively clear LWd values are uniformly lower than the opaquely cloudy values for each $T_{2m}$ (Fig. 6a). As netLW is the quantity used to discriminate between clear and cloudy points, this is expected. On the other hand, radiatively clear SWd values are higher than opaquely cloudy SWd values for each $\theta$ (Fig. 6b). This corresponds to the shortwave albedo effect, i.e. clouds reflect solar radiation back to space. The magnitude of this shortwave cloud albedo effect is variable, even for a fixed solar zenith angle. As a first order approximation, this variation is due to the cloud optical properties as the albedo varied little over the measurement period. Equation (7) reproduces the spread of observed values for $\tau_{SW}$ between 1.7 to 28.2, a range which is coherent with total column COD values from previous studies (Sect. 4.3). In contrast, the longwave warming effect (i.e., the difference between the dashed/dotted and solid lines in Fig. 6a) varies little either as a factor of $T_{2m}$ or $\tau_{LW}$, and remains close to $60 \, \mathrm{W \, m^{-2}}$.

COD variations therefore have a non-negligible impact on the surface radiative balance. For $\theta = 60°$, for example, there is an approximately $200 \, \mathrm{W \, m^{-2}}$ difference in SWd between the optically thinnest and thickest clouds. This translates into a total shortwave cloud forcing that ranges between $-20$ to $-60 \, \mathrm{W \, m^{-2}}$, assuming an albedo of 0.8 (typical of the N-ICE campaign April-June period). This range is significant when it is contrasted to the typical longwave forcing of $\approx 60 \, \mathrm{W \, m^{-2}}$: even for $\theta = 60°$, only the optically thickest clouds could contribute to cool the surface during the April-June N-ICE2015 campaign period. Most clouds continued to warm the surface. This is explored in more depth in Sect. 5.4.

### 5.4 Beyond N-ICE2015: estimating the summer cloud net radiative forcing at the surface

The parametrisations introduced in Sect. 5.3 appear to work well when confronted with N-ICE radiative flux data: for CODs between 1.8 and 27.8, Eq. 7 reproduces the observed spread of downwards shortwave flux values at each zenith angle (Fig. 6). They can therefore be used to study the cloud net radiative forcing at the surface (netCF) and its dependence on solar zenith angle, albedo, and cloud optical depth. netCF is calculated according to the following equations:

$$\mathrm{CF}_{SW} = (1 - \alpha) \cdot \mathrm{F}_0(\theta) \cdot (T_c(\theta, \tau_{SW}, \alpha) - 1)$$
$$\mathrm{CF}_{LW} \simeq 60 \, \mathrm{W \, m^{-2}}$$
$$\mathrm{netCF} = \mathrm{CF}_{SW} + \mathrm{CF}_{LW} \tag{8}$$

with $\mathrm{CF}_{SW}$ the cloud shortwave radiative forcing and $\mathrm{CF}_{LW}$ the cloud longwave radiative forcing. These are counted as positive if they contribute to warm the surface, and negative if they contribute to cool it. In practice, $\mathrm{CF}_{LW}$ is positive and $\mathrm{CF}_{SW}$ is negative. Because $\mathrm{CF}_{LW}$ appears to depend little on surface temperature (Sect. 5.3), it will be considered constant. $T_c$ and $F_0$ are the cloud broadband shortwave transmission and the clear-sky downwards shortwave radiative flux respectively, which are calculated as in Sect. 5.3.

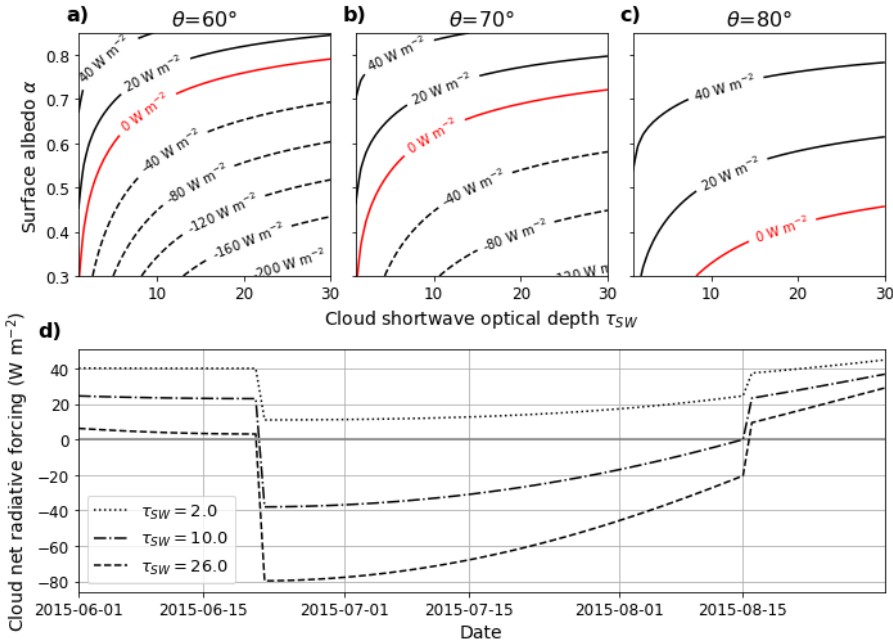

**Figure 7.** Panels a-c: iso-contours of net surface radiative forcing as a function of albedo and cloud shortwave optical depth for three different solar zenith angles (Eq. 8). Dashed black lines correspond to negative iso-contours, solid black lines to positive iso-contours, and red lines to the $0 \text{ W m}^{-2}$ iso-contour. Panel d: Calculated evolution of the net surface cloud radiative forcing, for three different CODs (dotted line: $\tau_{SW} = 2$; dash-dotted line: $\tau_{SW} = 10$; dashed line: $\tau_{SW} = 26$), over the 2015 summer period. The summer variation of the albedo is constructed based on values from the NCAR Climate System Model (Weatherly et al., 1998), and the solar zenith angle values are daily averages at 82°N, 14°W (approximate position of the N-ICE ice camp).

The output of Eq. 8 is shown in Fig. 7a-c for varying values of the surface albedo $\alpha$ and the cloud shortwave optical depth
$\tau_{SW}$ for zenith angle values $\theta = 60°$, 70° and 80°. For each angle, the evolution is the same: netCF increases with $\alpha$ and decreases with $\tau_{SW}$. Since $CF_{LW}$ is considered to be constant, this is a shortwave effect. Optically thick clouds reflect more shortwave radiation than optically thin clouds, and the magnitude of this shortwave radiative cooling is larger over low-albedo surfaces. Indeed, since high-albedo sea ice reflects most of the incoming radiation, clouds have a lower absolute impact on the radiative balance over these surfaces. The solar zenith angle affects netCF in a similar fashion. For given values of $\alpha$ and $\tau_{SW}$,
netCF increases with $\theta$. The red line in Fig. 7a-c represents the $0 \text{ W m}^{-2}$ iso-contour, and therefore delimits the regions of the ($\tau_{SW}$,$\alpha$) plane in which clouds have a total net radiative cooling or warming effect. The higher the solar zenith angle, the smaller the region of net radiative cooling.

Equation 8 can also be used to estimate a summer cycle of netCF beyond the end of the N-ICE campaign period. In order to do that, values of $\theta$ and $\alpha$ must be chosen. While $\theta$ is easily calculated for a given date and location (here 82°N, 14°W, which
is the approximate position of the N-ICE ice camp), $\alpha$ must be parametrised. We chose the four-level parametrisation for multiyear sea-ice used in the NCAR Climate System Model (Weatherly et al., 1998), which has been shown to agree well with

SHEBA data (Perovich, 2002). In this model, cold snow is considered to have an albedo of $0.82$, melting snow of $0.75$, melting ice $0.5$ and cold ice of $0.65$. The transition between different surface types is naturally dependent on the specific location and year, but an approximate cycle can be constructed. Here the surface is set to be melting snow up to 21 June, melting ice from 21 June to 15 August, and cold ice from 15 August onwards. Indeed, the measured albedo was $0.74$ (corresponding to melting snow) at the end of the N-ICE2015 campaign, i.e. on the 19 June 2015.

The results of this calculation are shown in Fig. 7d. Up to the 21 June 2015, only the optically thickest clouds ($\tau_{SW} = 26$) have a netCF which approaches zero, while optically thin clouds still contribute to warm the surface. This is in accordance to the N-ICE2015 measurements (Sect. 5.3). As the surface transitions from melting snow to melting ice on the 21 June, the netCF increases abruptly. This shows the important impact of $\alpha$ on the net cloud radiative forcing. However, $\tau_{SW}$ is almost as large a source of variability. The netCF for optically thin clouds ($\tau_{SW} = 2$) remains positive, i.e. they continue to warm the surface, while optically thick clouds ($\tau_{LW} = 26$) have a strong net surface cooling effect of $-80$ W m$^{-2}$. The netCF increases with the $\theta$, and netCF values become positive for all $\tau_{SW}$ values with the surface transition to cold ice on the 15 August.

This approximate calculation of summer netCF exhibits negative values from the end of June to early August. This is coherent with the previous studies in the central Arctic Ocean, which showed that clouds exerted a cooling effect (i.e. negative radiative forcing) on the surface from the end of June to July (Shupe et al., 2006). It is also coherent with the observation that during IAOOS, surface temperatures were lower in the absence of clouds for spring and autumn months, but not during the summer. However, netCF in these months also appears to depend strongly both on the surface and cloud type. Optically thin clouds may continue to warm the surface throughout the summer while thick, liquid water clouds will have a strong surface cooling effect. In considering the effect of clouds on the surface radiative balance during the summer, it is therefore important to have an accurate estimation of COD and surface albedo. This strong variability in summer netCF may also contribute to explain that the 2 m temperature of cloudless profiles during IAOOS was not different at statistically significant level from that of cloudy profiles in June, July and August (Sect 5.1). Indeed, if summer netCF over the central Arctic Ocean were uniformly negative, for all clouds, the surface should be observed to be colder in the presence than in the absence of clouds.

# 6 Conclusions

The IAOOS field campaign (2014 - 2019) consisted in the deployment of instrumented buoys in the Arctic sea ice. In this study, the whole IAOOS lidar dataset was treated and analysed. This included correcting for window frost as outlined in Mariage (2015) and deconvoluting the signal to reduce the effects of receiver saturation in bright conditions. An algorithm was implemented to detect cloud layers and calculate their optical depth, either directly when applicable or through the IAB by assuming a constant lidar ratio. Surface radiative flux data from the N-ICE campaign, during which four IAOOS buoys were deployed, and from ERA5 reanalyses, was also exploited.

The low number of profiles in some months causes some uncertainty on specific monthly cloud properties. However, the results show statistically significant differences in cloud cover and optical and geometrical properties of clouds between the summer and April, November and December. Low cloud cover (i.e., with a base beneath 2 km) is found to be $76\%$ averaged

over all months of the campaign. Monthly cloud frequency is minimum in April and November/December and over $85\%$ from May - October, with two small maxima in June and October. First-layer clouds are geometrically thickest in October, and thinnest in the summer. This is likely linked to moisture intrusions from the Atlantic in early autumn. Lastly, first-layer cloud bases are found to be extremely low in all seasons: under 120 m in a vast majority of cases.

The IAOOS lidar detects multiple cloud layers at much lower rates than other instruments, because the first cloud layer usually dampens the signal completely. Total cloud optical and geometrical thicknesses from previous campaigns and satellite data are much larger than those measured by IAOOS, especially in the summer when multilayered clouds are known to be most common. The single-layer COD as measured by IAOOS is highest in October.

The surface impact of Arctic clouds is also seasonally variable. In October and November, clouds warm the surface: 2 m temperatures associated with cloudless profiles are up to 8 K colder than those associated with profiles containing at least one low cloud. However, there is no statistically significant difference in surface temperatures between cloudless and cloudy profiles in the summer.

Data from the IAOOS lidar deployed during the N-ICE campaign allowed us to identify two modes in the N-ICE measured netLW distribution in late spring/summer. The "radiatively clear" netLW mode, centered around $-72$ W m$^{-2}$, is associated with cloudless IAOOS lidar profiles, while the "opaquely cloudy" mode is centered around $-11$ W m$^{-2}$ and is linked to cloudy lidar profiles. These are analogous to the well-known winter radiative modes, except that the opaquely cloudy mode is much more prevalent (over $80\%$) and that the two modes have a 60 W m$^{-2}$ difference, compared to 40 W m$^{-2}$ in the winter. Clouds exert a larger longwave warming in the summer than in the winter, probably linked to the higher proportion of liquid water in clouds. Clouds in the spring/summer also have a shortwave cooling effect. This is shown to depend not only on solar zenith angle and albedo, but also strongly on COD.

During the N-ICE2015 April to June period clouds were observed exert a positive radiative forcing on the surface, with the cloud shortwave albedo effect cancelling out its longwave warming effect only for very large optical depths at zenith angles $> 60°$. Over the full central Arctic Ocean summer cycle, it is estimated that optically thick clouds cause a negative radiative forcing of $-80$ W m$^{-2}$ but that optically thin clouds continue to have a warming effect. It is therefore important to have a good estimation of whole-column COD in order to calculate the radiative effect of clouds on the surface. The compensation of the cloud longwave warming effect by the shortwave cooling effect explains that there is no clear difference in near-surface temperature between IAOOS cloudless and cloudy profiles during the summer months.

The measured surface radiative fluxes were compared to the output of the ERA5 reanalyses. ERA5 does not accurately reproduce the observed bimodality of the spring/summer netLW distribution. Indeed, it does not correctly identify cloudless periods during April and May (but not June). This issue should be investigated.

Over the period 2014-2019, the IAOOS buoys have delivered 1777 lidar profiles. Despite technical difficulties with both the lidar and the data analysis, this campaign has offered a medium-term 3-season picture of the Arctic lower troposphere above 82°N from ground-based measurement, which is an important complement to satellite data. These results help to broaden our understanding of the Arctic low cloud cover and its impacts on the surface. However, more measurements would be needed to further characterise Arctic clouds. In particular, combined radiometer-radar-lidar measurements would be crucial to allow

the study of radiative impacts to be generalised to late summer and especially autumn, when clouds are optically thick and frequent.

*Data availability.* N–ICE2015 observational data sets are available from the Norwegian Polar Data Centre (https://data.npolar.no/dataset/) and are cited in the text (Hudson et al., 2016). IAOOS atmospheric data used in this paper are available upon request to the corresponding author and are available through the AERIS Data Portal at https://www.aeris-data.fr/. The ERA5 re–analysis products can be retrieved at
650 http://apps.ecmwf.int/.

## Appendix A:  Determination of a 90% confidence interval for cloud occurrence frequency

Let us suppose that the event "presence of a cloud with base $< 2$ km in a given IAOOS lidar profile" follows a Bernoulli distribution of parameter $p$, with $p$ the cloud frequency. This seems plausible given that the profiles are at least 6 hours apart, and the events can therefore considered to be independent. We aim to determine a confidence interval for $p$ based on:

1. Previous studies of clouds in the Arctic, which have shown that p is generally around $0.7$;

    2. The IAOOS measurements: for each month $m$, there are $n_m$ profiles of which $k_m$ contain at least one cloud with base $< 2$ km.

From 1), an "a priori" probability distribution for $p$ can be conceived: for example $\mathcal{N}(0.7, 0.15)$, normalised over the $[0,1]$ interval. Using the Bayes formula, the IAOOS measurements can then be taken into account to calculate an updated $\Pr(p|\text{meas})$
for each month $m$:

$$\Pr(p|\text{meas}) = \frac{\Pr(\text{meas}|p) \cdot \Pr(p)}{\Pr(\text{meas})} \tag{A1}$$

with

$$\Pr(\text{meas}|p) = \mathcal{B}(k_m; n_m, p)$$
$$\Pr(\text{meas}) = \sum_{p_i} \Pr(\text{meas}|p_i) \cdot \Pr(p_i)$$
$$= \sum_{p_i} \mathcal{B}(k_m; n_m, p_i) \cdot \Pr(p_i) \tag{A2}$$

where $p_i$ are the possible values of the parameter $p$, and $\mathcal{B}(k; n, p) = \binom{n}{k} p^k (1-p)^{n-k}$ is the binomial probability mass
function with parameters $n$ and $p$. The results of this calculation are shown in Fig. A1, which synthesises the results of Sect. 4.1: the probability distributions for the months of May - October show significant overlap. However, they do not overlap at all with the November, December, March and April distributions, although these are much wider because of the lower number of measurements.

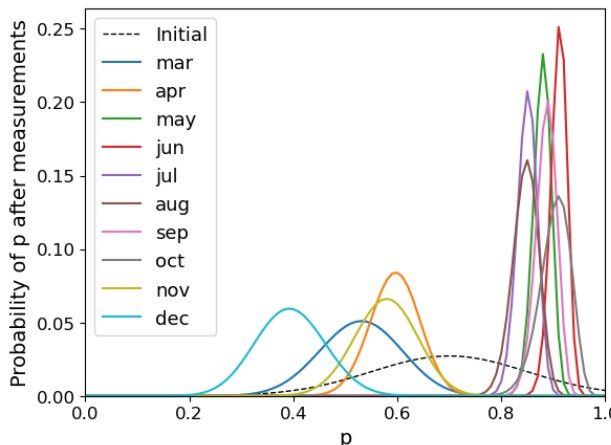

**Figure A1.** $\mathrm{Pr}(p|\mathrm{meas})$ as a function of the cloud occurrence frequency p. The dashed black line corresponds to the a priori distribution. The updated distributions for each month (Eq. A1) are shown in colour.

| Optical depth | 5th percentile | Median | 95th percentile |
|---|---|---|---|
| $\tau_{LW}$ | 1.4 | 2 | 2.5 |
| $\tau_{SW}$ | 1.2 | 7.8 | 20.2 |
| $\tau_{808}$ | 0.5 | 0.9 | 1.9 |

**Table B1.** Statistical range (5th, 50th and 95th percentiles) of three different estimations of optical depth: $\tau_{LW}$ (from the downwards longwave flux), $\tau_{SW}$ (from the downwards shortwave flux) and $\tau_{808}$ (calculated from the IAOOS lidar profiles). For a robust comparison, $\tau_{LW}$ and $\tau_{SW}$ values considered here are interpolated on the IAOOS profile times. The percentiles are therefore established over 54 data points which correspond to the 54 IAOOS profiles. Individual errors carried over from measurement errors on LWd, SWd and $T_{2m}$ are in the range $8 - 19\%$ (mean 11%) for $\tau_{SW}$, and $8 - 23\%$ (mean 13%) for $\tau_{LW}$.

The 5th and 95th percentiles of the distribution of $p$ determined through Eq. A1 can then be calculated to yield a 90% confidence interval.

## Appendix B: Contribution of the lowest cloud layer to the total column COD

Cloud optical depths measured by the IAOOS lidar correspond only to the lowest cloud layer, and not to the total column (Sect. 4.3). Here we attempt to evaluate the contribution of this lowest layer to the total column COD. This would allow better comparison of IAOOS CODs to existing satellite statistics. Furthermore, as seen in Sect. 5.3, total column shortwave COD is the quantity that most impacts the surface radiative balance. Equations 6 and 7 were inverted using a numerical equation solver to calculate the broadband shortwave and longwave CODs $\tau_{SW}$ and $\tau_{LW}$ from the N-ICE SWd, LWd and temperature values at the time of the IAOOS profiles. Albedo was taken as fixed and equal to $0.8$ in this calculation. The measurement errors of

SWd, LWd and temperature (Sect. 2.2.1) as well as the choice of a fixed albedo create an error on $\tau_{SW}$ and $\tau_{LW}$ which is estimated through a Monte Carlo method. This error is no more than $19\%$ for $\tau_{SW}$ and $23\%$ for $\tau_{LW}$ (Table B1).

In analysing the results, it must be taken into account that the longwave optical depth of any single cloud layer is smaller than its shortwave optical depth. The shortwave-to-longwave optical depth ratio depends on the microphysical properties of clouds (droplet phase, radius) and a precise determination would require the help of radiative transfer models. In this manner, Garnier et al. (2015) calculates $\tau_{532nm}/\tau_{12\mu m} \approx 1.8$ for ice particles with an effective diameter between $5$ and $60$ microns. We use this value as a rule of thumb to enable comparison between $\tau_{LW}$, $\tau_{SW}$ and the IAOOS optical depths $\tau_{808}$.

$90\%$ of $\tau_{LW}$ values obtained in this manner fall in the $1.4 - 2.5$ range (Table B1). It must be noted that these $\tau_{LW}$ values do not capture the optical depth of the whole column. Indeed, because cloud emissivity $\epsilon_c$ tends to 1 exponentially, high $\tau_{LW}$ values are likely to be underestimated. Instead, this $\tau_{LW}$ must be seen as the part of the cloud cover whose emitted radiation reaches the surface. Inverting Eq. (7) yields shortwave optical depths between $1.2$ and $20.2$, with a median of $7.8$. This range shows much higher values than that of $\tau_{LW}$, even when accounting for the longwave-to-shortwave ratio. This is because the

shortwave radiative flux is impacted by the whole cloud column, and not only the first few layers. IAOOS optical depths ($\tau_{808}$ in Table B1) are much lower than both $\tau_{LW}$ and $\tau_{SW}$, with $90\%$ of values between $0.5$ and $1.9$. In fact, the ratio $\tau_{808}/(1.8 \cdot \tau_{LW})$ has a median value of $0.22$ (range $0.15 - 0.43$), while $\tau_{808}/\tau_{SW}$ has a median value of $0.11$ (range $0.03 - 0.68$). This means that first-layer clouds measured by IAOOS contribute around a quarter of the optical depth of clouds which have a longwave radiative impact on the surface, and $11\%$ of the total cloud column.

While this value is low, it is coherent with the observation that SHEBA-measured total cloud thicknesses are up to 7 times higher than the IAOOS-measured first layer thickness (Sect. 4.2). Regardless of potential underestimations in IAOOS measurements, it strongly suggests that further cloud layers must be present at higher altitudes. Some of these, possibly cirrus clouds, would then have a shortwave but no longwave impact on the surface. Furthermore, visual inspection of the relative humidity (RH) and temperature profiles obtained through radiosonde measurements during N-ICE supports the idea that the IAOOS

lidar correctly identifies the first cloud layer and probably misses higher cloud layers. Indeed, strong temperature inversion and diminution of RH are most often present at the lidar-identified cloud top. Further inversions and high RH values are often present, marking higher altitude cloud layers that are invisible to the lidar.

*Author contributions.* JM performed the data treatment and analysis and prepared the manuscript. FR and JCR provided supervision, guidance and editing. JP led the IAOOS project and designed the lidar. VM constructed the lidar and treated its data.

*Competing interests.* The authors declare that they have no conflict of interest.

*Acknowledgements.* The authors acknowledge support from Stephen Hudson and Lana Cohen at the Norwegian Polar Institute and Von P. Walden at Washington State University for use of the N-ICE2015 dataset. They acknowledge their use of SHEBA data provided by NCAR/EOL under the sponsorship of the National Science Foundation. This work was supported by the Equipex IAOOS (Ice Atmosphere Ocean Observing System) (ANR-10-EQPX-32-01), and by funding from the ICE-ARC program from the European Union 7th Framework Programme grant number 603887. Computer analyses benefited from access to IDRIS HPC resources (GENCI allocation A007017141) and the IPSL mesoscale computing center (CICLAD: Calcul Intensif pour le CLimat, l'Atmosphère et la Dynamique). This publication contains modified Copernicus Climate Change Service Information (2020). Neither the European Commission nor the ECMWF are responsible for any use that may be made of the Copernicus information or data in this publication.

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
