# Peer review of "Characterisation and surface radiative impact of Arctic low clouds from the IAOOS field experiment"

_Atmospheric Chemistry and Physics, 2020_

## Referee Comment (RC1) · Anonymous Referee #1 · 18 Oct 2020

The authors present results of the IAOOS field experiment which took place from 2014 and 2019 in the central Arctic. They focus on lidar measurements which were performed on drifting buoys and analyse cloud occurrence and further cloud properties based on this data set. They also look into the radiative fluxes measured during the N-ICE campaign, where collocated buoy observations were made as well, and analyse the different radiative modes observed in this time period. Here, they compare the observations to ERA5 reanalysis data.

Enhancing cloud observations in the Arctic is crucial to better understand Arctic clouds, they radiative impact and their impact on the Arctic climate system. Especially in the harsh Arctic environment and especially in the Central Arctic, it is quite challenging to acquire such data. Using drifting buoys with such instrumentation is quite impressive. The authors discuss the challenges of such observations and also the limitations. However, also the retrieved data is limited and lacks spatial (as the authors mention) and temporal coverage. This needs to be kept in mind when analysing and discussing the data. A maximum of 4 lidar profiles per day cannot provide robust cloud statistics. However, this is how the authors sell the results. In particular, monthly statistics based on this data, in particular for those months where the number of profiles is even less than 100, don't seem to be reliable. At ground-based surface observatories, continuous observations can be performed. Assuming a typical 1-min resolution of lidar measurement, this would result in at least 1440 profiles per day. What if only 4 measurements, i.e. 4 random snapshots of clouds at one day, were available instead? Would they capture the cloud statistics based on the high-resolution data? Don't get me wrong: I think that this data set is of high value but the representativity needs to be critically discussed. This is partly done in the manuscript but needs to be enhanced. During the N-ICE campaign, a micropulse lidar (MPL) is available. I strongly suggest to also include a section showing the comparison between the results of the buoy lidar and the MPL. This would provide more insight in the representativeness of the buoy lidar cloud observations. Please find in addition my specific comments below.

Specific comments:

I 5: "Cloud frequency is globally at 75%...": unclear what globally means. please be more specific (which time period exactly, region).

17: "On the whole, the cloud cover is very low...". Misleading. Could be read as: Cloud cover (=cloud fraction) is low (=small). Rather use "Cloud base height is very low..."

Il 26 ff: You could also mention the results of Mioche, G., Jourdan, O., Ceccaldi, M., and Delanoë, J.: Variability of mixed-phase clouds in the Arctic with a focus on the Svalbard region: a study based on spaceborne active remote sensing, Atmos. Chem. Phys., 15, 2445–2461, https://doi.org/10.5194/acp-15-2445-2015, 2015

II 41: Concerning ground-based cloud observations, you could mention the studies by
Shupe, M. D., V. P. Walden, E. Eloranta, T. Uttal, J. R. Campbell, S. M. Starkweather, and M. Shiobara, 2011: Clouds at Arctic Atmospheric Observatories. Part I: Occurrence and Macrophysical Properties. J. Appl. Meteor. Climatol., 50, 626–644, https://doi.org/10.1175/2010JAMC2467.1.

**and**

Shupe, M. D., 2011: Clouds at Arctic Atmospheric Observatories. Part II: Thermodynamic Phase Characteristics. J. Appl. Meteor. Climatol., 50, 645–661, https://doi.org/10.1175/2010JAMC2468.1.

Also, be aware of the enhanced cloud observations at Ny-Ålesund:

Nomokonova, T., Ebell, K., Löhnert, U., Maturilli, M., Ritter, C., and O'Connor, E.: Statistics on clouds and their relation to thermodynamic conditions at Ny-Ålesund using ground-based sensor synergy, Atmos. Chem. Phys., 19, 4105–4126, https://doi.org/10.5194/acp-19-4105-2019, 2019.

Ebell, K., T. Nomokonova, M. Maturilli, and C. Ritter, 2020: Radiative Effect of Clouds at Ny-Ålesund, Svalbard, as Inferred from Ground-Based Remote Sensing Observations. J. Appl. Meteor. Climatol., 59, 3–22, https://doi.org/10.1175/JAMC-D-19-0080.1.

II 45 ff: concerning shipborne and airborne observations it is also worth mentioning the ACLOUD and PASCAL campaigns:

Wendisch, M., and Coauthors, 2019: The Arctic Cloud Puzzle: Using ACLOUD/PASCAL Multiplatform Observations to Unravel the Role of Clouds and Aerosol Particles in Arctic Amplification. Bull. Amer. Meteor. Soc., 100, 841–871, https://doi.org/10.1175/BAMS-D-18-0072.1.

I 64: "extract a 5-year statistics of the Arctic cloud cover": This is overstated. It has to be made clear that this is not a robust statistic with respect to spatial and temporal coverage. Be more precise here: e.g. "cloud cover along the track of the drifting buoys in the central Arctic for the months of..."
I 89: As mentioned before, having only a maximum of four lidar measurements per day is a very, very low number. The representativity needs to be critically discussed. Not only once, but also when presenting the results.

I 101: "red line": ambiguous, better "red circle"

II 106: information on N-ICE campaign:

Please include more information about the campaign data set and the auxiliary instrumentation, e.g. detailed information about the radiation sensors ("four component radiometer"). What kind of instruments exactly? What are the instrument specifications? I assume they both down- and upward radiative fluxes are provided, right? Where is the instrumentation exactly installed? Distance of the instruments to each other?

Why is the information of the MPL not used in addition? The measurements of the MPL should be set in to context to the buoy lidar observations.

I 111: "April to June", please add 2015

I 114: please provide a reference for ERA-5

I 116: "L1"? Please be more specific

Il 122 ff: How does the lidar window frost impact cloud detection? You could state at the end of section 3.1.1 what this means for the accuracy of the cloud observations?

I 160: Equation 1: please introduce all variables!

I 174: How is the threshold of 1.1 chosen? What is the impact on cloud detection?

I 190 Tc has not been introduced

I 192: Equation 2: make sure that all variables are introduced, e.g. alpha\_p

I 224: "Global" is misleading: Why not simply name it as it is: "average monthly cloud cover from March to December"

**ACPD**
Il 228 ff: just a comment here: low clouds frequently occur in the Arctic and it is especially difficult for satellites to capture these clouds also from active instrumentation, e.g. due to blind zone, ground clutter. Ground-based observations are thus crucial to capture these low clouds. It is true that for ground-based observations the sensitivity is highest in the lower atmosphere but the combination of cloud radar and lidar can very well capture the whole atmospheric column!  $\rightarrow$  (Il 235-236).

II 236: which instruments were used in the Hahn et al study?

Il 240-251 and Fig.2: You really need to discuss your results in conjunction with the number of measurements (as seen also in Table 2). The very low cloud occurrence in March, April, November and December is very suspicious. I would not overinterpret the results here. Please consider the representativity of the data. Discussion of "Interannual variability": I would also be careful here. I am not convinced that based on the number of data, any conclusions can be drawn here.

I 256: "lacks spatial coverage": but also temporal coverage! Again, a comparison between MPL data and the buoy lidar data are crucial to give more confidence in the temporal representativity of the results. You really need to draw your conclusions more carefully.

Table 2: Are the numbers in each months are for all buoy drifts/years?

I 270: "non-significant": did you perform a significance test?

I 277: This is a too strong statement. Please rewrite.

I 281: "(note however...)". Thank you that you mentioned that point here but not sufficiently discussed and highlighted.

I 286: "significant difference": tested?

I 287: "shoulder months": unclear, please do not use "shoulder months" throughout the manuscript and mention the months explicitly.

ACPD
I 300: Again, how representative are the 222 profiles?

Fig. 4. "a" and "b" missing in plot Do you calculate the median and percentiles from e.g. 6 values? See for example March

I 337 ff: I am not convinced that you can simply set the COD to 2 for high-IAB cloud layers. You simply do not know the COD in these cases. You state that this is helpful for examining the seasonal trend. But also this trend has large uncertainties then.

II 382-383: Why was the information from the MPL not exploited as well?

1 389: "...due to the higher surface temperatures in spring/summer." Please elaborate on that.

Fig.5 b) and c) Remove "Measured" in xlabel since also ERA5 data are shown Explain RC1, RC2, OC in figure caption. Rather provide a detailed section on where the radiation sensors are installed in the text than mentioning it in the figure caption.

I 404, Equation 5 What about the surface emissivity? Should be included in equation 5.

I 420: "...partly compensated by a 14 Wm-2 error in LWu in April/May..." Where can I see this? It would be interesting to include a plot of LWu.

Figure 6: Please use different line styles for the different cloud optical depth. Remove "Evolution" from figure caption: "Longwave downward radiative flux as a function of ..."

II 425: Which kind of satellite data are assimilated in ERA5 exactly? Please provide more details here which underline your hypothesis.

II 441 ff: It is totally unclear why you need to come up with parameterizations or estimates of the downward radiative flux components. These are measured, aren't they? What is the intention of this part?

I 473: "shortwave cloud albedo effect"
II 476-478: "In contrast, the longwave warming effect,...." Maybe it would be good to remind the reader that this is the difference between the dashed and solid line in Fig.6 (as far as I understood).

II 482-484: "This explains that..." Can you elaborate on that a little bit more? Unclear to me.

II 489-491: "Equations 6 and 7 were inverted to calculate..." Can you explain in detail how you did it? In Eq. 7, COD is not directly included. It might be good to remind the reader how this is connected to transmittance. Do you take F0 from your fitted function?

Table 4: What are the uncertainties of the derived COD values?

Il 521-522 and this section: "The results show a significant seasonal variation...". Overstated due to the reasons mentioned before. Also "Monthly cloud frequency is minimum in March/April and November/December..." A discussion of measurement and sampling uncertainties is needed here! I doubt that the results a robust for these months.

**ACPD**

---

## Referee Comment (RC2) · Anonymous Referee #2 · 22 Oct 2020

Arctic low clouds are a key climate feature of the atmospheric boundary layer over the Arctic Ocean. Arctic low clouds are important because of their strong influence on the amount of solar and infrared radiation that is incident on the surface. In the meantime, they can strongly modify the low-level heat, moisture and momentum fluxes. This paper quantified the seasonality and surface radiative impacts of Arctic low clouds from the Ice, Atmosphere, Arctic Ocean Observing System (IAOOS) field campaign. It is a very important topic as the Arctic is a data-sparse region. Moreover, both passive and active remote sensing products have their limitations on polar cloud retrievals. Therefore, the information obtained from this five-year campaign is very valuable. Overall, this paper is well written, but the structure needs to be improved. I recommend it to be accepted after following issues being addressed. Please find my specific concern as below.

[Figure]

Overall: The current version contains too much information. I find it a bit difficult to follow because of the paper's structure, which is not well organized and logical. The section 4.1.4 is tightly connected with section 4.3. The author also mentioned that "The reasons for this are explored in Sect. 4.3 by investigating the summer radiative balance." (line 362-363). Is it better to combine these two sections together? From my perspective, a better structure would be the seasonality of cloud properties, impact of cloud on surface temperature and radiation budget, and followed by the comparison of ERA5 to surface in-situ measurements. And I am quite sure how to combine section 4.4 with other sections. Also, I believe the authors need to add transitional sentences and paragraphs to connect these sections in a more logical way. Line 5-6: "Cloud frequency is globally at 75%, and above 85% from May to October." Why the cloud frequency is globally? Not in the Arctic? Line 59-60: I think you could also mention that CALIPSO satellite product has limitation on temporal coverage, which is only available after 2006. Figure 4: There are no (a) and (b) in the figures. Section 4.1.4 and Table 3: How many cloudy and cloudless profiles are there for each moth? For example, you may rarely get cloudless profiles in summer as low cloud frequency is pretty high. Does this issue affect your results? Section 4.1.4: The clear-sky LW flux also exerts large influence on surface temperature. In most of cases, the magnitude of clear-sky LW flux is larger than that of cloud longwave radiative effect. We usually believe that the high pressure tends to reduce clouds and associated cloud warming effect. However, the high pressure in the upper troposphere could also increase the clear-sky LW flux and enhance surface warming. In addition, the authors tried to investigate the impacts of clouds on surface temperature by using lidar profiles with and without low clouds. Then how to make sure other conditions (e.g. large-scale circulation) remain same between two groups? I understand that this may not easy to be addressed. But authors should treat this issue more carefully. Reference: Ding, Q., Schweiger, A., L'Heureux, M., Battisti, D. S., Po-Chedley, S., Johnson, N. C., ... & Steig, E. J. (2017). Influence of high-latitude atmospheric circulation changes on summertime Arctic sea ice. Nature Climate Change, 7(4), 289-295. Line 425: "This may ultimately

be due to an error in the satellite data that is assimilated by the ERA5 reanalyses."
Which satellite data is assimilated by the ERA5? Can you be more specific about this
bias? Line 464-467: Is N-ICE second period from April to June? Since you used a
fixed surface albedo 0.8, which excludes the impacts of reduced multiple reflections
between surface and clouds with sea ice melt, particularly from April to June. Can you
comment on that? Line 480: "This translates into a total shortwave cloud forcing that
ranges between $-20$ to $-60$ W m$-2$, assuming an albedo of 0.8." Again, I believe
that surface albedo plays an important role in determining the shortwave flux at the
surface. Assuming a surface albedo of 0.8 could totally ignore the multiple reflections
between clouds and melting surface. Reference: Wendler, G., Moore, B., Hartmann,
B., Stuefer, M., & Flint, R. (2004). Effects of multiple reflection and albedo on the
net radiation in the pack ice zones of Antarctica. Journal of Geophysical Research:
Atmospheres, 109(D6). Line 522: "Low cloud cover (i.e., with a base beneath 2 km) is
found to be 76% globally over the course of the campaign." What it is globally?

Please also note the supplement to this comment:
https://acp.copernicus.org/preprints/acp-2020-918/acp-2020-918-RC2-
supplement.pdf

---

## Author Comment (AC1) · 15 Dec 2020

*Replies to both referee comments are contained in this document. Referee comments are in black, and our responses in blue. Modifications to the manuscript are in italics, with the line number (for the revised manscript) in bold.*

**Reply to RC1**

**Global comments :**
The authors present results of the IAOOS field experiment which took place from 2014and 2019 in the central Arctic. They focus on lidar measurements which were performed on drifting buoys and analyse cloud occurrence and further cloud propertiesbased on this data set. They also look into the radiative fluxes measured during theN-ICE campaign, where collocated buoy observations were made as well, and analyse the different radiative modes observed in this time period. Here, they compare the observations to ERA5 reanalysis data.

Enhancing cloud observations in the Arctic is crucial to better understand Arctic clouds,they radiative impact and their impact on the Arctic climate system. Especially in the harsh Arctic environment and especially in the Central Arctic, it is quite challengingto acquire such data. Using drifting buoys with such instrumentation is quite impressive. The authors discuss the challenges of such observations and also the limitations.
However, also the retrieved data is limited and lacks spatial (as the authors mention) and temporal coverage. This needs to be kept in mind when analysing and discussing the data. A maximum of 4 lidar profiles per day cannot provide robust cloud statistics. However, this is how the authors sell the results. In particular, monthly statistics based on this data, in particular for those months where the number of profiles is even less than 100, don't seem to be reliable. At ground-based surface observatories, continuous observations can be performed. Assuming a typical 1-min resolution of lidar measurement, this would result in at least 1440 profiles per day. What if only 4 measurements, i.e. 4 random snapshots of clouds at one day, were available instead? Would they capture the cloud statistics based on the high-resolution data? Don't get me wrong: I think that this data set is of high value but the representativity needs to be critically discussed. This is partly done in the manuscript but needs to be enhanced.

During the N-ICE campaign, a micropulse lidar (MPL) is available. I strongly suggest to also include a section showing the comparison between the results of the buoy lidar and the MPL. This would provide more insight in the representativeness of the buoy lidar cloud observations. Please find in addition my specific comments below.

First of all, we would like to thank the reviewer for his comments, which are both detailed and pertinent, as well as for the suggested references. We are sensible of the time and effort which must have been spent in reviewing the manuscript in this way.

Firstly, we will address the reviewer's main points, which we understand to be :
- that the statistics lack robustness due to the low number of points, especially in the months of October, November, December, March and April ;
- that a maximum of 4 profiles/day may not be representative of the higher-resolution cloud statistics ;
- and that therefore the robustness and representativity of the IAOOS lidar profiles must be critically discussed, in order to avoid overstating the results. The reviewer suggests that we compare our data with that of the MPL available during the N-ICE campaign.

We fully agree that a critical discussion of these points is necessary.

Robustness
The number of lidar profiles yielded by the IAOOS buoys is naturally lower than that at a ground based station, because the buoys are autonomous and therefore have a limited power supply and no supervision by an operator. In particular, as noted by the reviewer, there were less than 100 profiles/month for the October – April period. This is due to the especially harsh winter conditions. It naturally creates uncertainty on the calculated monthly cloud statistics. However, the seasonal variability appears robust, i.e. distributions of cloud properties do in general differ at a stastically significant level between summer and April, November, and December.

Here, we will discuss the cloud occurrence frequency in particular, as the reviewer notes in his specific comments that the low cloud frequencies in the months of November/December and March/April are suspicious, and we believe this to be an important result of the paper.

Putting aside the number of profiles for the moment, the reviewer's comments indicate that the obtained cloud frequencies (56 % in November, 32 % in December, 46 % in May and 59 % in April) are inherently suspicious because they are too low. We are surprised by this assessment. With the exception of December, the obtained values are within the envelope of previous studies, even from ground-based observations (Shupe et al, 2011 ; Wang and Key, 2004 ; Zygmuntowska, 2012 ).

In order to make our case more rigorously, we have calculated 90 % confidence intervals on the obtained monthly cloud occurrence frequencies. This is done in the following way, for each month :

- We suppose that the event "presence of a cloud with base<2 km in a given IAOOS lidar profile" has a probability p, with p the cloud frequency. Since the profiles are at least 6h apart, the events can be considered to be independent. Then, the number of profiles in each month which contain at least one cloud follows a binomial distribution with parameters p and n (n is the total number of profiles in a month).

- The probability distribution of p, taking into account our monthly measurements, can then be calculated from an a priori distribution using the Bayes formula. In practice, the a priori distribution doesn't have a great impact ; based on values found in the literature, we have chosen a normal distribution centered on 0.7.

- The 5th and 95th percentiles of this probability distribution can be calculated to obtain a 90 % confidence interval for p (e.g., 29 % - 51 % for December).

This method is further explained in Appendix A to the revised manuscript.

The probability distributions for the months of May to October show significant overlap (Fig. 1). However, they do not overlap at all with the November, December, March and April distributions, although these are much wider because of the lower number of measurements. We contend that the IAOOS measurements do therefore show significant seasonal variability in cloud frequency between winter and summer, with a transition between October and November and April and May.

For other cloud quantities we have applied the Mann-Whitney U test to show that there are statistically significant differences between months (see revised manuscript).

[Figure]

Fig. 1 : Monthly probability distributions of p after the measurements are taken into account (the dashed line is the a priori distribution, which is the same for each month).

Representativity
The other main difficulty expressed by the reviewer pertained to the representativity of the IAOOS time sampling. The choice was made for the IAOOS campaign to have the lidar shoot only every six hours and sometimes less, depending on conditions and the need to preserve battery power. This was of course partly for practical reasons – it would have been impossible for the lidar to shoot every minute. But it is also part of the approach. Similar to worldwide radiosonde launches which occur every 12 hours, one lidar profile every six hours is assumed to represent a random sample of cloud conditions over the buoy operation period.

This approach can, and should, be discussed.
1) Shooting at a given time only four times/day may also induce a bias in the resulting statistics if there is a strong diurnal variability in cloud cover. To give an extreme example : a sample of profiles acquired at 3 UTC and 9 UTC would obviously not be representative if clouds only ever occurred between 3:30 UTC and 8:30 UTC.
2) Whether or not one profile every six hours is enough to represent cloud statistics naturally depends on the timescale of cloud variations. For another extreme example, there would be no point in sampling at a 1 minute resolution if clouds maintained themselves for months at a time.

To answer the first point : in general, there is little diurnal cycle at high latitudes, since the shortwave radiation varies more on a seasonal than on a daily scale. This is supported by the literature. Analysing data from six Arctic ground stations, Shupe et al. (2011) found that on average, the cloud occurrence anomaly from the daily mean was less than 5 percentage points. Our opinion is that  the lidar sampling at constant times does not induce any strong bias in the statistics.

As to the second point, the 6-12h sampling timestep must be compared to the timescales of cloud variability in the Arctic. Shupe et al. (2011) analysed the temporal persistence of cloud layers in the Arctic from data at six Arctic ground stations. They found that median cloud persistence ranges between 3.1 to 4.5h among the studied sites, while the mean varies from 8 to 22h. These values are of the same order as the IAOOS time step, so we expect that the sampling should not produce large errors.

We generated random series of alternating cloudy/clear periods from lognormal distributions which respect the means and medians of Shupe et al. These series each had a total length of 744h, or 31 days. The mean cloud occurrence frequency (COF) over each random series was then calculated using different sampling timesteps, from

1 min to 48h. The results were compared to the COF calculated with a timestep of 1 min.

Over a total of 300 such random series, the highest absolute error (top 5th percentile) incurred on COF by a sampling timestep of 12h was around 8 %. For a timestep of 6h, this value was only 5 %. In fact, it appears that it is not necessary to sample at very high temporal resolution in order to get a good picture of the overall cloud occurrence frequency.

The reviewer suggested comparing the IAOOS lidar data with the N-ICE MPL, which had a 1-min timestep, to see if a 6h sampling timestep correctly reproduces the cloud statistics. This is an interesting idea, but nothing guarantees that such a case study would in turn be representative. We believe the above explanation is quantitatively more robust.

**Specific comments**:

l 5: "Cloud frequency is globally at 75%...": unclear what globally means. please be more specific (which time period exactly, region).
« Globally » here means the April – December average over the whole campaign period. The text has been edited to make this clearer and now reads :

*(l 5) The average cloud frequency from April to December over the course of the campaign was 75%. Cloud occurrence frequencies were above 85% from May to October.*

l7: "On the whole, the cloud cover is very low...". Misleading. Could be read as: Cloudcover (=cloud fraction) is low (=small). Rather use "Cloud base height is very low..."
This has been edited accordingly.

ll 26 ff: You could also mention the results of Mioche, G., Jourdan, O., Ceccaldi, M.,and Delanoë, J.: Variability of mixed-phase clouds in the Arctic with a focus on the Svalbard region: a study based on spaceborne active remote sensing, Atmos. Chem.Phys., 15, 2445–2461, https://doi.org/10.5194/acp-15-2445-2015, 2015

ll 41: Concerning ground-based cloud observations, you could mention the studies by

Shupe, M. D., V. P. Walden, E. Eloranta, T. Uttal, J. R. Campbell, S. M. Starkweather,and M. Shiobara, 2011: Clouds at Arctic Atmospheric Observatories. Part I: Oc-currence and Macrophysical Properties. J. Appl. Meteor. Climatol., 50, 626–644, https://doi.org/10.1175/2010JAMC2467.1.

and

Shupe, M. D., 2011: Clouds at Arctic Atmospheric Observatories. Part II: Ther-modynamic Phase Characteristics.J. Appl.Meteor.Climatol., 50, 645–661,https://doi.org/10.1175/2010JAMC2468.1.

Also, be aware of the enhanced cloud observations at Ny-Ålesund:

Nomokonova, T., Ebell, K., Löhnert, U., Maturilli, M., Ritter, C., and O'Connor,E.: Statistics on clouds and their relation to thermodynamic conditions at Ny-Ålesund using ground-based sensor synergy, Atmos. Chem. Phys., 19, 4105–4126, https://doi.org/10.5194/acp-19-4105-2019, 2019.

Ebell, K., T. Nomokonova, M. Maturilli, and C. Ritter, 2020: Radiative Effect of Clouds at Ny-Ålesund, Svalbard, as Inferred from Ground-Based Remote Sensing Observations.J. Appl. Meteor. Climatol., 59, 3–22, https://doi.org/10.1175/JAMC-D-19-0080.1.

ll 45 ff: concerning shipborne and airborne observations it is also worth mentioning the ACLOUD and PASCAL campaigns:

Wendisch, M., and Coauthors, 2019: The Arctic Cloud Puzzle: Using ACLOUD/PASCAL Multiplatform Observations to Unravel the Role of Clouds and Aerosol Particles in Arctic Amplification. Bull. Amer. Meteor. Soc., 100, 841–871, https://doi.org/10.1175/BAMS-D-18-0072.1.

Thank you for the references. They have been cited in the revised manuscript.

l 64: "extract a 5-year statistics of the Arctic cloud cover": This is overstated. It hast o be made clear that this is not a robust statistic with respect to spatial and temporal coverage. Be more precise here: e.g. "cloud cover along the track of the drifting buoys in the central Arctic for the months of . . ."
We agree that the shortcut used led to an overstatement of the scope of the dataset. We have amended this sentence to:

*(l 66) ...to extract a multi year statistic of the April to December cloud cover along the track of the drifting buoys.*

l 89: As mentioned before, having only a maximum of four lidar measurements per day is a very, very low number. The representativity needs to be critically discussed. Not only once, but also when presenting the results.
This point has been discussed in the general answer above. Having only four profiles a day was partly due to measurement constraints, but it was also a methodological choise as one lidar profile every six hours is assumed to represent a random sample of cloud conditions. Indeed, because clouds persist on average from 8h to 22h, it is not necessary to have one profile per minute in order to measure monthly values of cloud occurrence frequency.

l 101: "red line": ambiguous, better "red circle"
This has been edited accordingly.

ll 106: information on N-ICE campaign: Please include more information about the campaign data set and the auxiliary instrum entation, e.g. detailed information about the radiation sensors ("four component radiometer"). What kind of instruments exactly? What are the instrument specifications ? I assume they both down- and upward radiative fluxes are provided, right? Where is the instrumentation exactly installed? Distance of the instruments to each other? Why is the information of the MPL not used in addition? The measurements of the MPL should be set in to context to the buoy lidar observations.
We have added more information about the campaign dataset and the instruments. This paragraph now reads :

*(l 109) The Norwegian Young Sea Ice Experiment (N-ICE) campaign took place from January to June 2015. During that time, theresearch vessel Lance drifted with four different ice floes (Walden et al.; Cohen et al., 2017; Walden et al., 2017). The first two drifts took place during the winter (January - March 2015) while the last two drifts occurred in the late spring to early summer period (April to June 2015). On each floe, a "Supersite" ice camp was installed about 300m away from the research vessel. Atmospheric measurements were mostly performed at this Supersite. Surface*

*longwave fluxes (up and down) were measured with a Kipp & Zonen CGR4 pyrgeometer, which has a 4.5 to 42 µm bandwith. The shortwave fluxes (up and down) were measured with a Kipp & Zonen CMP22 pyranometer (200 to 3600 nm bandwidth). Both these instruments were heated and ventilated using a Kipp & Zonen CVF4 unit. Their accuracy is 3% (or 5 W m$^{-2}$) for the shortwave, and 2% (or 3 W m$^{-2}$) for the longwave (Walden et al., 2017; Hudson et al., 2016). The temperature at two meters was measured with a ventilated and shielded Vaisala HMP-155A sensor which has an accuracy of 2.4% (or 0.3°C) (Graham et al., 2017; Cohen et al., 2017). In addition, radiosondes were launched twice-daily from the research vessel, yielding profiles of relative humidity, temperature and wind speed (Walden et al., 2017).*

*Four IAOOS buoys were deployed during this campaign and drifted in the ice floe close to the research vessel. In particular, the B12 buoy was locked into the third ice floe 200m away from the Supersite from end of April to the beginning of June 2015 (Fig. 1). Because of the proximity of the buoy to the Supersite over this period, the N-ICE surface radiative flux and temperature measurements can be used as a complement to the IAOOS data. This allowed us to evaluate the radiative impact of clouds on the surface in late spring to early summer (Sect. 5.2.1 and 5.3).*

Our reasons for not using the MPL as a comparison to the lidar buoy has been discussed above.

l 111: "April to June", please add 2015
This section has been modified as shown in the above paragraph.

l 114: please provide a reference for ERA-5
The following reference has been added :

*Hersbach, H., Bell, B., Berrisford, P., Hirahara, S., Horányi, A., Muñoz-Sabater, J., Nicolas, J., Peubey, C., Radu, R., Schepers, D., Simmons,A., Soci, C., Abdalla, S., Abellan, X., Balsamo, G., Bechtold, P., Biavati, G., Bidlot, J., Bonavita, M., Chiara, G., Dahlgren, P., Dee,D., Diamantakis, M., Dragani, R., Flemming, J., Forbes, R., Fuentes, M., Geer, A., Haimberger, L., Healy, S., Hogan, R. J., Hólm, E.,620Janisková, M., Keeley, S., Laloyaux, P., Lopez, P., Lupu, C., Radnoti, G., Rosnay, P., Rozum, I., Vamborg, F., Villaume, S., and Thépaut, J.-N.: The ERA5 global reanalysis, Quarterly Journal of the Royal Meteorological Society, 146, 1999–2049,* https://doi.org/10.1002/qj.3803,2020*.*

l 116: "L1"? Please be more specific
The L1 marked a bibliography reference to the link for acquiring ERA5 data on the ECMWF website. This was indeed quite unclear, and it has been updated to the following :
*(l 129) (Copernicus Climate Change Service (C3S), 2017)*

ll 122 ff: How does the lidar window frost impact cloud detection? You could state at the end of section 3.1.1 what this means for the accuracy of the cloud observations?
As noted in this section, the frost modifies the system constant C (by lowering the window transmission). Once the modified C is determined using the method outlined in this section, it is plugged in to the attenuated scattering ratio (SRatt) calculation. The obtained SRatt should therefore be independent of window frost and there should be no further impact on cloud detection.

Of course, the modified C determined through this method may be slightly off. Mariage (2015) estimates that the error on C is around 30 % for a frost index between 0,1 and 0,3. The impact of cloud detection then depends on the sign of the error. If C is erroneously high, for example, SRatt will be erroneously low. This would lead to features being harder to detect (and therefore, feature bases being too high). On the

other hand, if C is erroneously low, SRatt will be erroneously high and spurious features may be detected.

In practice, visual inspection of the profiles indicates that the cloud detection algorithm is robust to the errors caused by the window frost correction. However, it is difficult to quantify the impact on the cloud observations.

We have added the following comments at the end of section 3.1.1 :
**(l 151)** *[...] this frost correction method naturally causes uncertainty on the obtained value of C. Around 11% of profiles have values of γ between 0.1 and 0.3. In this case, Mariage (2015) estimates that the window frost correction leads to a 30% error onC. A further 3%of profiles have 0.05≤γ <0.1, in which case the error on C can be up to 60%. For γ≥0.3, the C error tends towards the frost-free system constant determination error, which is around 10% (Mariage, 2015). The system constant is used in the calculation of the attenuated scattering ratio,from which all cloud quantities are derived (Sect. 3.2). However, it is difficult to quantify the impact of its error on cloud detection, in part because it depends on the sign of the error. An overestimated C would lead to under-detection of cloud layers, and vice versa. In practice, visual inspection of the profiles indicates that the cloud detection algorithm outlined below is robust to the errors that may be incurred during the window frost correction.*

l 160: Equation 1: please introduce all variables!
This has been corrected.

l 174: How is the threshold of 1.1 chosen? What is the impact on cloud detection?
In the absence of clouds (or aerosols) the attenuated scattering ratio (SRatt) should be 1 near the ground. Setting the threshold at 1.1 allows for a 10 % margin to avoid small fluctuations of the  system constant C from « triggering » the algorithm into detecting a feature where there is none.

Setting the threshold lower therefore risks detecting spurious features. Setting the threshold higher would make it harder to detect a feature base. The algorithm would then risk either overestimating the altitude of the feature base, or missing the feature altogether (if it is thin). From visual inspection of the profiles, 1.1 appeared to be a good compromise.

Note that cloud layers, as opposed to aerosols, have large SRatt values. At the cloud base, SRatt increases very rapidly to values often >100. In practice, therefore, the specific value of the threshold (from 1.1 to 1.5, for example) has little impact on detection of cloud layers or on the determination of their base.

l 190 Tc has not been introduced
The text has been amended to introduce Tc (which is the cloud transmission).

l 192: Equation 2: make sure that all variables are introduced, e.g. alpha_p
Thank you for catching this. alpha_p was introduced right after Equation 2.

l 224: "Global" is misleading: Why not simply name it as it is: "average monthly cloud cover from March to December"
The suggested change was made.

ll 228 ff: just a comment here: low clouds frequently occur in the Arctic and it is especially difficult for satellites to capture these clouds also from active instrumentation,e.g. due to blind zone, ground clutter. Ground-based observations are thus crucial to capture these low clouds. It is true that for ground-based observations

the sensitivity is highest in the lower atmosphere but the combination of cloud radar and lidar can very well capture the whole atmospheric column! →(ll 235-236).
We thank the reviewer for this comment.

ll 236: which instruments were used in the Hahn et al study?
The Hahn et al study relies on surface weather reports from ships and ice camps, i.e. visual inspection of the sky. Naturally, these are quite uncertain in the dark, and the object of the study is to account for this issue by introducing a nighttime correction factor.

The text has been updated to note this point:
**(l 261)** *Averaging visual observations from ships and ice-camps, Hahn et al [...]*

ll 240-251 and Fig.2: You really need to discuss your results in conjunction with the number of measurements (as seen also in Table 2). The very low cloud occurrence in March, April, November and December is very suspicious. I would not overinterpret the results here. Please consider the representativity of the data. Discussion of "Inter-annual variability": I would also be careful here. I am not convinced that based on the number of data, any conclusions can be drawn here.
This point has been discussed in the general answer. We agree with the reviewer, however, that these paragraphs required further justification and that some conclusions should have been more careful.
We decided to eliminate references to March data throughout the manuscript, since this month had less than 30 profiles. We have changed ll 240-255 to the following paragraphs to clarify our reasoning and modulate our conclusions :

**(l 266)** *The results of IAOOS dataset are shown in Table 3 and Fig. 2. Note here that the number of profiles available for each month is variable, both because of the more favorable operation conditions in the summer and the timing of the buoy deployment (usually in May). As such, there are more than 200 profiles from May to September, around 100 in April and October, and less than 54 in November and December. Months with less than 30 profiles, i.e. January, February, and March, are not treated in this article. Care must therefore be taken in analysing the results of late autumn and winter. A 90% confidence interval for the cloud occurrence frequency can be estimated from a Bayesian calculation, assuming that the number of cloudy profiles followsa binomial distribution and supposing an appropriate a priori distribution for the cloud frequency from the literature (Appendix A).*

*The IAOOS data shows a similar trend as the literature, with generally higher cloud cover values. From May to October, clouds are present over 85% of the time (Fig. 2). In contrast to the previous ground-based climatologies outlined above, there are two peaks at more than 0.9 in the monthly cloud frequency, although they differ little from the summer baseline. The first is in June, which has a mean cloud frequency of 0.92 and a confidence interval of (0.88−0.94). The second peak is in October, also with a mean cloud frequency of 0.92 but with a slightly wider confidence interval (0.85−0.95) because of the lower number of profiles. This is reminiscent of the results of Zygmuntowska et al. (2012), from CALIPSO data, which show a peak in cloud occurrence above 0.9 in October. July and August have slightly lower cloud frequency values (0.85 (0.82−0.88) and 0.85 (0.8−0.89) respectively). However, since there is non negligible overlap between the confidence intervals of June/October and the other summer months, it is difficult to draw solid conclusions as to May - October variability.*

*In the IAOOS dataset, April and November appear to mark a sharp transition in cloud occurrence frequency from the summer values. April has a cloud frequency of 0.59 (0.52−0.67) while the cloud frequency in November is 0.56 (0.48−0.68). While the confidence intervals are quite wide here due to the lower number of profiles, there is*

*no overlap with the summer confidence intervals. This suggests that the lower cloud frequencies observed during the months of April and November is meaningfully different from that of the months of May through October. December cloud frequency is lower still, at 0.32 (0.29−0.51). Note however the width of the confidence interval and the fact that the December data corresponds to a single year of measurement (2017).*

*It is not possible to robustly quantify interannual variability in Arctic cloud cover from the IAOOS dataset since there are at most four years of data for each month. Qualitatively, however, the April - May transition in cloud frequency observed by the buoys is quite variable. In 2014, the B02 buoy observed a very sharp spring transition in cloud frequency: from 40% (35%,60%) in April 2014 to more than 90% (89%,97%) in May and June 2014 (blue circles, Fig. 2). On the other hand, this transition was much more gradual in 2017 (buoy B24, orange diamonds). The June 2017 cloud frequency is less than 80% (69%,85%), overlapping significantly with the May 2017 cloud frequency confidence interval of (56%,78%). This is not an effect of spatial variability as both B02 and B24 were drifting in the Atlantic sector of the Arctic (Fig. 1).*

l 256: "lacks spatial coverage": but also temporal coverage! Again, a comparison between MPL data and the buoy lidar data are crucial to give more confidence in the temporal representativity of the results. You really need to draw your conclusions more carefully.
The part on spatial variability was simplified further and clarified. Our answer as to temporal representativity was given in the general comments.

*(l 297) It has been observed from satellite data that the Atlantic sector is the cloudiest part of the Arctic Ocean (Liu et al., 2012; Wang and Key, 2004). This is linked to the low pressure systems and the storm tracks arriving from the northern Atlantic Ocean. Since most of the IAOOS buoys drifted in this sector, the IAOOS dataset must be regarded as most representative of these specific conditions, and not of the ocean-wide cloud characteristics.*

Table 2: Are the numbers in each months are for all buoy drifts/years?
Yes. The caption has been edited to specify this :

*[...] N_p is the total number of lidar profiles for each month* **(for all years and buoys)**.

l 270: "non-significant": did you perform a significance test?
No. We used « non-significant » here in its meaning of « insignificant », i.e. small. We acknowledge that this makes the statement confusing as « non-significant » in this context would most likely be interpreted in its statistical meaning.
However, the statistical significance question is interesting. Using Fisher's exact test, the November multilayered cloud occurrence can be shown to be different from the July value at a statistically significant level (p-value = 0.007 for a two-sided alternative hypothesis) - despite the low number of profiles.

This has been reworded to :

*(l 313) Only one IAOOS profile contained multilayered clouds in November, and none in December. Despite the low number of total profiles in these months, these values are different from the July multilayered cloud frequency at a statistically significant level; for November, Fisher's exact test yields a p-value of 0.007(Fisher, 1922).*

l 277: This is a too strong statement. Please rewrite.
This has been rewritten to :

*(**l 323**) Clouds in the IAOOS dataset are extremely low, with little seasonal variability [...]*

l 281: "(note however...)". Thank you that you mentioned that point here but not sufficiently discussed and highlighted.
The two sentences preceding l 281, i.e.

« [...] In March, only 57% of cloud bases are below 120 m. Another 29% of first layer cloud bases are between 120 and 500 m, which still corresponds to low level, likely boundary layer clouds (note however the low number of profiles in this month). »

have been removed, as we have decided not to treat the March data due to the low number of points.

l 286: "significant difference": tested?
Yes. The Mann-Whitney U test (for difference between the means of two independent samples) gives the following results :
- July (n1 = 355) and October (n2 = 104) : U=9834.5, p-value < 0.001
- July (n1 = 355) and April (n2 = 60) : U = 5940.5, p-value < 0.001
This has been added to the manuscript :

*(**l 333**) This difference appears significant at a statistical level. The Mann-Whitney U for the July and October cloud thickness distributions was 9834.5 (with sample sizes n1=355 and n2=104), yielding a p-value<0.001 (Mann and Whitney, 1947). The same is true for July and April (U= 5940.5, n1= 355 and n2= 60, p-value<0.001).*

l 287: "shoulder months": unclear, please do not use "shoulder months" throughout the manuscript and mention the months explicitly.
We agree that this expression is unclear and have removed it from the manuscript.

l 300: Again, how representative are the 222 profiles?
The calculation of S* from 222 profiles is not truly a scientific result of the paper, but more of a methodological point. Another option for calculating COD would have been to use a constant value of S* drawn from the literature. However, we felt it would be more robust to draw S* from our dataset itself, where possible. As it appeared that median S* values from our dataset vary substantially between months, we further felt that simply using the median value from the 222 profiles would bias the results, and so decided to use monthly S* values.

The object of this paragraph is to confront our values of S* with the literature, to check that they are within the enveloppe of expected values. As noted, we do not have any data about the microphysical composition of the clouds and so it is impossible to draw conclusions about the reasons for the variations of S*. This is why we have not looked further into the representativity of these 222 profiles. Such work would be interesting, but outside of the scope of this paper.

As noted before, we have decided not to include the March data in this paper. We have therefore deleted the following sentences, starting l. 311 :
*« For example, the very high values observed in March [...] independent of other seasonal or temperature effect. »*

Fig. 4. "a" and "b" missing in plot. Do you calculate the median and percentiles from e.g. 6 values? See for example March

« a » and « b » have been added to the plot, thank you for catching this. The median and percentiles for the lidar ratio are calculated from the points shown on the plot for each month. The March data is no longer included but the reviewer's point is applicable to April and November/December, which have less than 15 points each. As discussed above, we chose to use these monthly median values of S* for the calculation of COD despite the low number of points.

l 337 ff: I am not convinced that you can simply set the COD to 2 for high-IAB cloud layers. You simply do not know the COD in these cases. You state that this is helpful for examining the seasonal trend. But also this trend has large uncertainties then.
The reasoning for setting the COD at 2 for high-IAB cloud layers is the following :
- high-IAB cloud layers are, as a group, expected to have higher COD than low-IAB cloud layers ;
- low-IAB cloud layers have a 95th percentile value of 2
Therefore, for calculation of the median only, it makes some sense to set COD for high-IAB cloud layers to 2. In the median calculation, it ensures that the fact that high-IAB cloud layers exist, and are expected to have higher COD than low-IAB cloud layers, is accounted for.

The reviewer is correct that the ensuing trend is not certain. The alternative would be simply to set aside the high-IAB layers and treat only the low-IAB cases (this is the filled line in Fig. 4). But this would be akin to showing only half of the picture. We chose to show both calculations in order to avoid giving a false impression of the seasonal trend ; the peak in October appears robust, but the values in June (and generally in the summer) are quite probably larger than the values yielded by the low-IAB calculations.

We feel that this is a more rigorous presentation of our results. However, this was not explained in a clear way in the present manuscript. We have made the following changes :

*(l 381) To overcome this problem, the COD of high-IAB cloud layers was set to 2. This value was chosen as it is the 95th percentile of CODs calculated for low-IAB layers, and high-IAB cloud layers are as a group expected to have higher COD than low-IAB layers. The monthly median COD was then calculated including these high-IAB cases (Fig. 4, filled squares). This correction is not quantitatively robust as the value of 2 is arbitrarily chosen, not calculated. However, it accounts for the fact that high-IAB cloud layers exist, and are expected to have higher COD than low-IAB cloud layers, in the calculation of the median. This is helpful for examining the seasonal trend, which otherwise is biased by the presence of noise.*

*It creates a significant difference in June and July, the months in which the percentage of high-IAB cloud layers is the highest. With this correction, the median monthly COD exhibits two peaks (June and October) and a minima in April. The October peak is however still the annual maximum, and does not appear to be strongly impacted by the inclusion of high-IAB cloud layers. Previous satellite measurements have exhibited a pattern of higher COD in spring and autumn, for instance May and October for the AVHRR data (Wang and Key, 2004) over the Arctic Ocean. The IAOOS dataset exhibits this October peak in single-layer COD. Another peak in June appears possible, although the IAOOS measurements are very uncertain in this month.*

ll 382-383: Why was the information from the MPL not exploited as well?
The spirit of this study consists in extracting a multiyear statistic from the IAOOS database. Although the N-ICE2015 offers a more complete dataset than the IAOOS lidar from April – June 2015, it only covers three months of one year and does not therefore add meaningfully to the statistic.

The reviewer suggests using the MPL to check that the IAOOS lidar is able to reproduce its higher-resolution statistics. However, as stated in the general answer, we do not believe that such a case study would necessarily be representative.

l 389: "...due to the higher surface temperatures in spring/summer." Please elaborate on that.
This sentence is unclear and would indeed require more justification; for example, the impact of surface temperature on the downwards as well as the upwards flux. We have chosen to simplify it to the following observation:

*(l 454) […] the netLW mode values are lower than in the winter. Indeed, both the downwards and upwards components of the longwave flux (LWd and LWu) increase from winter to summer. However, LWu increases more than LWd in both modes, causing a shift to lower netLW values.*

Fig.5 b) and c) Remove "Measured" in xlabel since also ERA5 data are shown. Explain RC1, RC2, OC in figure caption. Rather provide a detailed section on where the radiation sensors are installed in the text than mentioning it in the figure caption.
« Measured » has been removed. The caption now reads :

*Panel a: time series of surface net longwave measurements during the N-ICE field experiment (second period, April-June 2015). The vertical lines indicate the time of IAOOS lidar profiles, with red lines corresponding to cloudless profiles. Panels b and c: histogram of the measured (filled line) and ERA5 (dashed line) net longwave flux during the N-ICE winter (b) and spring/summer (c) campaign periods. Panels d and e: hourly ERA5 vs measured net longwave in during the N-ICE winter (d) and spring/summer (e) campaign periods, with red dashed line indicating the 1:1 line. The colour corresponds to point density as calculated by a Gaussian kernel. For panel (e), three zones have been outlined. Zone "OC" contains points belonging to the opaquely cloudy mode of the measured netLW distribution. Zones "RC1" and "RC2" contain points belonging the radiatively clear mode of the distribution in April and May (RC1) and June (RC2).*

The location of the buoy and the radiation sensors is now described in Sect. 2.2.1, as per our answer to one of the reviewer's previous comments.

l 404, Equation 5 What about the surface emissivity? Should be included in equation 5.
Thanks for catching this. The surface emissivity should indeed be included. Walden et al (2017) suggest a value of 0.99 is appropriate for the N-ICE campaign. The following line has been added below the equation:

*(l 471) […] with epsilon the surface emissivity, which is assumed to be 0.99 (Walden et al., 2017).*

l 420: "...partly compensated by a 14 Wm-2 error in LWu in April/May..." Where can I see this? It would be interesting to include a plot of Lwu.
There is no plot of LWu in the paper as it stands. We discuss the relevant findings as to LWu and LWd in the text, for example in this sentence. Although a more detailed discussion of the upwards and downwards components of the flux would be interesting, perhaps they would best belong to another paper, focusing specifically on ERA5.

Figure 6: Please use different line styles for the different cloud optical depth. Remove "Evolution" from figure caption: "Longwave downward radiative flux as a function of..."
The suggested changes have been made.

ll 425: Which kind of satellite data are assimilated in ERA5 exactly? Please provide more details here which underline your hypothesis.

Infrared and microwave radiances from several different satellites are assimilated in ERA5. This includes measurements of cloud liquid water from the AMSR-2 instrument aboard GCOM-W1, AMSR-E aboard AQUA, GMI aboard the GPM Core Observatory, and others (see the ECMWF website : https://confluence.ecmwf.int/pages/viewpage.action?pageId=82870405#ERA5:datadocumentation-Table14)

However, we have decided to remove this hypothesis as it was not formulated on sufficiently solid grounds. ll. 425 to the end of the paragraph (in the original manuscript) has been replaced with a simple observation:

*(l 492) More investigation is required as to the ultimate source of this error.*

ll 441 ff: It is totally unclear why you need to come up with parameterizations or esti-mates of the downward radiative flux components. These are measured, aren't they? What is the intention of this part?

The goal of the parametrisation was to help discuss the impact of COD, SZA, and surface temperature on the downwards surface fluxes. This is mainly important for COD, which was not measured explicitly during SHEBA. It is also helpful for discussing the cloud net radiative forcing beyond the N-ICE campaign. This last part has been expanded in the modified manuscript as a response to the comments below.

l 473: "shortwave cloud albedo effect"
The suggested change has been made.

ll 476-478: "In contrast, the longwave warming effect, . . . ." Maybe it would be good to remind the reader that this is the difference between the dashed and solid line in Fig.6 (as far as I understood).

The reviewer's understanding is correct – and the suggested clarification is likely a good idea. The sentence has been edited to :

*(l 544) In contrast, the longwave warming effect (i.e., the difference between the dashed/dotted and solid lines in Fig. 6a) varies little [...]*

ll 482-484: "This explains that . . ." Can you elaborate on that a little bit more? Unclear to me.

We decided that this point would be best adressed in a new section (Sect. 5.4 « Beyond NICE2015 : estimating the summer cloud net radiative forcing at the surface » in the revised manuscript, **l 554 - 599**), in which we calculate and plot the net cloud radiative forcing directly.

ll 489-491: "Equations 6 and 7 were inverted to calculate . . ." Can you explain in detail how you did it? In Eq. 7, COD is not directly included. It might be good to remindthe reader how this is connected to transmittance. Do you take F0 from your fitted function?

In Eq. 7, LWd is calculated using
1) $F_0$ , which we fitted from the data and depends only on solar zenith angle;
2) T_c , which is calculated using the parametrisation of Fitzpatrick et al., 2003.
The parametrisation of T_c depends on solar zenith angle (theta), surface (albedo) alpha, and COD. For given values of theta, alpha and LWd we were therefore able to compute COD using a numerical equation solver (fsolve from the scipy.optimize package in python). We chose not to explicitly write out the parametrisation of Fitzpatrick in our paper because it is quite unwieldy, however every detail is in the referenced article.

Section 4.4 was moved to supplementary materials (Appendix B) in order to streamline the structure of the paper. However, we made the following modification :

*(l 674) Equations 6 and 7 were inverted **using a numerical equation solver** to calculate [...]*

Table 4: What are the uncertainties of the derived COD values?
There are three different sources of uncertainty for the derived COD values: a numerical uncertainty (from the solver), an uncertainty linked to the input parameter error (temperature, SWd, albedo and LWd), and the uncertainty due to the model itself. The first is expected to be small. The third is large but somewhat besides the point, as the aim here is specifically to compare IAOOS CODs to those obtained from inverting these models. Therefore we chose to focus on the second.

For tau_SW, the error ranged 8 % to 19 % (mean 11%). This was calculated by a Monte Carlo method. We drew 100 random values of albedo and SWd from the following distributions :
- for albedo : a normal distribution centered on 0.8 with a standard deviation of 0.025, therefore respecting the spread of actual measured albedos during N-ICE2015 (0.75 – 0.85);
- for SWd : a normal distribution centered on each measured SWd value and with a standard deviation equal to half the measurement error. This is the maximum of 3 % of the measured value and 5 W m$^{-2}$ (see presentation of the N-ICE measurements).

For tau_LW, the error ranged from  9 % to 23% (mean 13%). This was also calculated with a Monte Carlo method. 100 random values of temperature and LWd were drawn from the following distributions :
- for temperature : a normal distribution centered on each measured temperature value and with a standard deviation equal to half the measurement error. This is the maximum of 2.4 % of the measured value and 0.3°C (see presentation of the N-ICE measurements).
- for LWd : same as above, the measurement error is the maximum of 2 % of the measured value and 3 W m$^{-2}$.

In both cases, the error was then calculated as the mean absolute percentage error of the result over these 100 points.

This was specified in the caption to Table B1 :
*Individual errors carried over from measurement errors on LWd, SWd and T2m are in the range 8−22% (mean 14%) for $\tau_{SW}$, and 10−30% (mean 15%) for $\tau_{LW}$.*

And in the text:
*(l 674) Equations 6 and 7 were inverted using a numerical equation solver to calculate the broadband shortwave and longwave CODs $\tau_{SW}$ and $\tau_{LW}$ from the N-ICE SWd, LWd and temperature values at the time of the IAOOS profiles. Albedo was taken as fixed and equal to 0.8 in this calculation. The measurement errors of SWd, LWd and temperature (Sect. 2.2.1) as well as the choice of a fixed albedo create an error on $\tau_{SW}$ and $\tau_{LW}$ which is estimated through a Monte Carlo method. This error is no more than 19% for $\tau_{SW}$ and 23% for $\tau_{LW}$ (Table B1).*

ll 521-522 and this section: "The results show a significant seasonal variation...". Overstated due to the reasons mentioned before. Also "Monthly cloud frequency is minimum in March/April and November/December..." A discussion of measurement and sampling uncertainties is needed here! I doubt that the results a robust for these months.

*This has been reworded :*

*(l 607) The low number of profiles in some months causes some uncertainty on specific monthly cloud properties. However, the results show statistically significant differences in cloud cover and optical and geometrical properties of clouds between the summer and April, November and December.*
* * *
**Reply to RC2**

**Global comment**
Arctic low clouds are a key climate feature of the atmospheric boundary layer over the Arctic Ocean. Arctic low clouds are important because of their strong influence on the amount of solar and infrared radiation that is incident on the surface. In the meantime, they can strongly modify the low-level heat, moisture and momentum fluxes. This paper quantified the seasonality and surface radiative impacts of Arctic low clouds from the Ice, Atmosphere, Arctic Ocean Observing System (IAOOS) field campaign. It is a very important topic as the Arctic is a data-sparse region. Moreover, both passive and active remote sensing products have their limitations on polar cloud retrievals. Therefore, the information obtained from this five-year campaign is very valuable. Overall, this paper is well written, but the structure needs to be improved. I recommend it to be accepted after following issues being addressed. Please find my specific concern as below.

*We would like to thank the referee for the positive appreciation of our work and for the helpful suggestions below as to the structure.*

**Specific comments**
Overall: The current version contains too much information. I find it a bit difficult to follow because of the paper's structure, which is not well organized and logical. The section 4.1.4 is tightly connected with section 4.3. The author also mentioned that "The reasons for this are explored in Sect. 4.3 by investigating the summer radiative balance." (line 362-363). Is it better to combine these two sections together? From my perspective, a better structure would be the seasonality of cloud properties, impact of cloud on surface temperature and radiation budget, and followed by the comparison of ERA5 to surface in-situ measurements. And I am quite sure how to combine section 4.4 with other sections. Also, I believe the authors need to add transitional sentences and paragraphs to connect these sections in a more logical way.

*We agree that the paper has quite a complex structure, which might lead to confusion. We decided to take the reviewer's comment into account to make the paper's logical progression clearer.*
*- Sections 1, 2, 3 are unchanged*
*- Section 4 has been changed to « Seasonality of Arctic low clouds properties during IAOOS», covering the previous sections 4.1.1, 4.1.2, 4.1.3*
*- A new section 5, « Cloud impact on surface temperatures and radiative balance», covers previous sections 4.1.4, 4.2 and 4.3.*
*- A new section 5.4 « Beyond N-ICE2015: estimating the summer cloud net radiative forcing at the surface » was added discussing the cloud net radiative forcing in the summer.*
*- Section 4.4 has been moved to the appendix (Appendix B)*
*However, the section concerning ERA5 was kept just after the analysis of the two radiative modes. This is because it is not a global comparison of ERA5 to surface in-situ measurements over the course of the compaign, but a short evaluation of the*

representation of these modes in ERA5. We therefore feel that it makes more sense to keep these two sections in close connection.

The introduction has been updated to make the logical progression of the paper clearer, and we have added more transitional sentences (especially in Section 5). For example, at the end of Section 5.1 :

*(l 396) In the following sections, we look at the summer surface radiative balance in order to gain a better understanding of the mechanisms behind this seasonal variation in temperature difference between cloudy and cloudless profiles. First, the link between the net surface longwave flux and the presence of clouds is investigated (Sect. 5.2.1) from compared N-ICE and IAOOS measurements. Then, the influence of other factors such as solar zenith angle, temperature and COD on downwards shortwave and longwave fluxes during the N-ICE2015 April to June period is explored (Sect. 5.3). Lastly, the discussion of the net cloud radiative forcing at the surface is extended to the months of July and August using a simple parametrisation (Sect. 5.4)*

See also the beginning of Sect. 5.3 :
*(l 494) In the Arctic summer, clouds impact the surface radiative budget in two competing ways: they have a longwave warming effect and a shortwave cooling effect. In Sect. 5.2.1, the N-ICE2015 April-June netLW distribution was shown to be bimodal, withthe first mode corresponding to the presence of clouds in the IAOOS profiles and the second to their absence. However, other factors than the absence or presence of clouds may impact the surface radiative fluxes, both shortwave and longwave. In this section, the influence of variables such as the solar zenith angle, COD and surface temperature on the downwards fluxes (both longwave and shortwave) from the N-ICE2015 April-June period is explored and parametrisations of these fluxes are introduced.*

Line 5-6: "Cloud frequency is globally at 75%, and above 85% from May to October." Why the cloud frequency is globally? Not in the Arctic?
« Globally » here means the April – December average over the whole campaign period. The text has been edited to make this clearer and now reads :

*(l 5) The average cloud frequency from April to December over the course of the campaign was 75%. Cloud occurrence frequencies were above 85% from May to October.*

Line 59-60: I think you could also mention that CALIPSO satellite product has limitation on temporal coverage, which is only available after 2006.
This has been added, thank you for the suggestion.

*(l 62) Their record is also more limited in time than that of ground-based stations (from 2006 for CALIPSO)*

Figure 4: There are no (a) and (b) in the figures.
Thank you for catching this, it has been fixed.

Section 4.1.4 and Table 3: How many cloudy and cloudless profiles are there for each moth? For example, you may rarely get cloudless profiles in summer as low cloud frequency is pretty high. Does this issue affect your results?
The total number of profiles is indicated in Tables 2 and 3 ; we have not included the number of cloudy profiles in these tables but they can be calculated using the cloud fraction (which is indicated). The issue of the reliability of the statistics due to the low number of profiles in some months was raised in detail by the other referee, and we have decided to introduce confidence intervals to make the discussion of the results

more rigorous (see new Sect. 4.1). In light of these intervals we do not believe that the number of profiles affects our main point – which is that there is a statistically significant seasonal variability in cloud occurrence frequency. Please note that we have decided to restrict ourselves to months with more than 30 profiles, as this is the usual rule of thumb in statistics. All references to March statistics have therefore been deleted from the text.

See **II 266** onwards in the revised manuscript :
*The results of IAOOS dataset are shown in Table 3 and Fig. 2. Note here that the number of profiles available for each month is variable, both because of the more favorable operation conditions in the summer and the timing of the buoy deployment (usually in May). As such, there are more than 200 profiles from May to September, around 100 in April and October, and less than 54 in November and December. Months with less than 30 profiles, i.e. January, February, and March, are not treated in this article. Care must therefore be taken in analysing the results of late autumn and winter. A 90% confidence interval for the cloud occurrence frequency can be estimated from a Bayesian calculation, assuming that the number of cloudy profiles followsa binomial distribution and supposing an appropriate a priori distribution for the cloud frequency from the literature (Appendix A).*

*The IAOOS data shows a similar trend as the literature, with generally higher cloud cover values. From May to October, clouds are present over 85% of the time (Fig. 2). In contrast to the previous ground-based climatologies outlined above, there are two peaks at more than 0.9 in the monthly cloud frequency, although they differ little from the summer baseline. The first is in June, which has a mean cloud frequency of 0.92 and a confidence interval of (0.88−0.94). The second peak is in October, also with a mean cloud frequency of 0.92 but with a slightly wider confidence interval (0.85−0.95) because of the lower number of profiles. This is reminiscent of the results of Zygmuntowska et al. (2012), from CALIPSO data, which show a peak in cloud occurrence above 0.9 in October. July and August have slightly lower cloud frequency values (0.85 (0.82−0.88) and 0.85 (0.8−0.89) respectively). However, since there is non negligible overlap between the confidence intervals of June/October and the other summer months, it is difficult to draw solid conclusions as to May - October variability.*

*In the IAOOS dataset, April and November appear to mark a sharp transition in cloud occurrence frequency from the summer values. April has a cloud frequency of 0.59 (0.52−0.67) while the cloud frequency in November is 0.56 (0.48−0.68). While the confidence intervals are quite wide here due to the lower number of profiles, there is no overlap with the summer confidence intervals. This suggests that the lower cloud frequencies observed during the months of April and November is meaningfully different from that of the months of May through October. December cloud frequency is lower still, at 0.32 (0.29−0.51). Note however the width of the confidence interval and the fact that the December data corresponds to a single year of measurement (2017).*

Section 4.1.4: The clear-sky LW flux also exerts large influence on surface temperature. In most of cases, the magnitude of clear-sky LW flux is larger than that of cloud longwave radiative effect. We usually believe that the high pressure tends to reduce clouds and associated cloud warming effect. However, the high pressure in the upper troposphere could also increase the clear-sky LW flux and enhance surface warming. In addition, the authors tried to investigate the impacts of clouds on surface temperature by using lidar profiles with and without low clouds. Then how to make sure other conditions (e.g. large-scale circulation) remain same between two groups? I understand that this may not easy to be addressed. But authors should treat this issue more carefully.

Reference:

Ding, Q., Schweiger, A., L'Heureux, M., Battisti, D. S., Po-Chedley, S., Johnson, N. C., … & Steig, E. J. (2017). Influence of high-latitude atmospheric circulation changes on summertime Arctic sea ice. Nature Climate Change, 7(4), 289-295.

If we understand correctly, the reviewer's argument is the following :
- The downwards longwave flux (LWd) is the sum of a clear sky component (LWd$_{cs}$) and of a clouds component (LWd$_{cl}$) – if clouds are present.
- LWd$_{cs}$ is expected to vary according to synoptic conditions. In particular, high pressures (in the upper levels of the troposphere) are associated with subsidence, warming the troposphere and therefore increasing LWd$_{cs}$ (Ding et al, 2017.)
- High pressures (at which level?) are also associated with less clouds.
- The magnitude of variation of LWd$_{cs}$ is comparable or larger than the cloud longwave radiative effect (LWd$_{cl}$).
- Therefore, total LWd becomes :
  - LWd = LWd$_{cs}$(1) under high pressures (no cloud effect, high LWd$_{cs}$)
  - LWd = LWd$_{cs}$(2) + LWd$_{cl}$ under low pressures (clouds, but low LWd$_{cs}$)
and (LWd$_{cs}$(1) – LWd$_{cs}$(2)) ~ LWd$_{cl}$, which means that there is little total difference between LWd under high pressures (cloudless conditions) and low pressures (cloudy conditions).
Therefore, the absence of an observed surface temperature difference between cloudy and cloudless profiles could simply be due to a compensating effect in clear sky LWd.

This is an interesting and valid point. As a first remark, this mechanism would also be expected to hold true in autumn and spring : however, we do observe a significant surface temperature difference between cloudy and cloudless profiles in these seasons. Secondly, this is only schematic. The different variables and their variations would need to be quantified. For example, at what frequency do clouds occur under high and low pressure respectively ? (For that matter, is the confounding variable surface pressure or geopotential in the higher levels of the troposphere?) What is the magnitude of variation of LWd$_{cs}$ in comparison with LWd$_{cl}$ ? Without these informations, it is hard to tell if the proposed effect would be significant or not.

While Ding et al (2017) establish a link between increased geopotential at 200 hPa and increased LWd and temperature at the surface, they do not distinguish between clear sky and cloud LWd. In contradiction with the proposed mechanism above, they show that higher geopotential at 200 hPa is linked to a decrease in mid and high-level clouds and a slight increase in low-level cloudiness over the central Arctic Ocean. They judge this to be consistent with the observed augmentation in LWd.
It is not clear therefore that the above mechanism would be valid, as « higher pressures » (i.e. higher geopotentials at 200 hPa) are not associated with less clouds.

However, it is true that large-scale circulation is an important parameter that we fail to control for. As pointed out by the reviewer and outlined above, a true treatment of this issue would be complex and out of the scope of this paper.
Some elements of an answer are below.
The IAOOS buoys were equipped with barometers as well as temperature sensors. It appears that surface pressures for « cloudy » and « cloudless » profiles are not different at a statistically significant level except in August and November. In both of these months, the lidar profiles that contain clouds appear to coincide with markedly higher surface pressures than those that don't contain clouds (+12 hPa). However, there is a strong temperature difference in November but not in August. In all other summer months cloudy and cloudless profiles appear to have similar surface pressures. In short the two groups do not appear to sample wildly different conditions.

This has been clarified in the manuscript.

*(l 407) As noted before, louds are naturally not the only factor impacting surface temperatures or even the downwards longwaver adiative flux. Large-scale circulation is also important: for example, high geopotential at 200hPa is linked to a warming of the troposphere through subsidence, which increases the longwave radiative flux received at the surface (Ding et al., 2017). It is therefore important to check that cloudy and cloudless lidar profiles do not sample different surface pressures. The IAOOS buoys were equipped with barometers as well as temperature sensors. It appears that surface pressures for cloudy and cloudless profiles are not different at a statistically significant level, with the exception of August and November. In both of these months, the lidar profiles that contain clouds appear to coincide with markedly higher surface pressures than thosethat don't contain clouds (+12 hPa, Mann-Whitney test p-values <0.005). As surface temperatures in the two groupes differ strongly in November but not in August, however, surface pressure does not appear to be a confounding factor for surface temperature and cloud occurrence.*

Line 425: "This may ultimately be due to an error in the satellite data that is assimilated by the ERA5 reanalyses." Which satellite data is assimilated by the ERA5? Can you be more specific about this bias?
Infrared and microwave radiances from several different satellites are assimilated in ERA5. This includes measurements of cloud liquid water from the AMSR-2 instrument aboard GCOM-W1, AMSR-E aboard AQUA, GMI aboard the GPM Core Observatory, and others (see the ECMWF website : https://confluence.ecmwf.int/pages/viewpage.action?pageId=82870405#ERA5:datadocumentation-Table14)

However, we have decided to remove this hypothesis as it was not formulated on sufficiently solid grounds.  ll. 425 to the end of the paragraph has been replaced with a simple observation:

*(l 492) More investigation is required as to the ultimate source of this error.*

Line 464-467: Is N-ICE second period from April to June? Since you used a fixed surface albedo 0.8, which excludes the impacts of reduced multiple reflections between surface and clouds with sea ice melt, particularly from April to June. Can you comment on that?
Yes, the second period is from April to June. We used a fixed albedo of 0.8 because in practice, the measured albedo during the N-ICE April-June period varied only from 0.75 – 0.84, which doesn't change the LWd much using the Fitzpatrick parametrisation. For example, for a solar zenith angle of 55° and a cloud optical depth of 20, the difference in LWd between an albedo of 0.75 and 0.8 is only 7 %. As our model is mainly illustrative, this is an acceptable error.

Line 480: "This translates into a total shortwave cloud forcing that ranges between −20 to −60 W m −2 , assuming an albedo of 0.8." Again, I believe that surface albedo plays an important role in determining the shortwave flux at the surface. Assuming a surface albedo of 0.8 could totally ignore the multiple reflections between clouds and melting surface.

Reference:
Wendler, G., Moore, B., Hartmann, B., Stuefer, M., & Flint, R. (2004). Effects of multiple reflection and albedo on the net radiation in the pack ice zones of Antarctica. Journal of Geophysical Research: Atmospheres, 109(D6).

We agree with the reviewer that the albedo impacts the shortwave flux quite consequently. The assumption is only in regards to the N-ICE dataset, for which the

albedo did not vary much from a value in 0.8. The following phrase has been added for clarification:

**(l 549)** *[...] assuming an albedo of 0.8* **(typical of the N-ICE campaign April-June period)**

Please note that in response to this comment and another comment made by the first reviewer, we have decided to add a new Section 5.4. « Beyond NICE2015 : estimating the summer cloud net radiative forcing at the surface » which explores the impact of albedos on this parametrisation.

Line 522: "Low cloud cover (i.e., with a base beneath 2 km) is found to be 76% globally over the course of the campaign." What it is globally?
This has been modified to

**(l 609)** *averaged over all months of the campaign*

*for more clarity.*